# VARIATIONAL AUTOENCODER WITH ARBITRARY CONDITIONING

**Oleg Ivanov**
Samsung AI Center Moscow
Moscow, Russia
tigvarts@gmail.com

**Michael Figurnov**
National Research University
Higher School of Economics *
Moscow, Russia
michael@figurnov.ru

**Dmitry Vetrov**
Samsung-HSE Laboratory,
National Research University
Higher School of Economics
Samsung AI Center Moscow
Moscow, Russia
vetrovd@yandex.ru

## ABSTRACT

We propose a single neural probabilistic model based on variational autoencoder that can be conditioned on an arbitrary subset of observed features and then sample the remaining features in "one shot". The features may be both real-valued and categorical. Training of the model is performed by stochastic variational Bayes. The experimental evaluation on synthetic data, as well as feature imputation and image inpainting problems, shows the effectiveness of the proposed approach and diversity of the generated samples.

## 1 INTRODUCTION

In past years, a number of generative probabilistic models based on neural networks have been proposed. The most popular approaches include variational autoencoder (Kingma & Welling, 2013) (VAE) and generative adversarial net (Goodfellow et al., 2014) (GANs). They learn a distribution over objects $p(x)$ and allow sampling from this distribution.

In many cases, we are interested in learning a conditional distribution $p(x|y)$. For instance, if $x$ is an image of a face, $y$ could be the characteristics describing the face (are glasses present or not; length of hair, etc.) Conditional variational autoencoder (Sohn et al., 2015) and conditional generative adversarial nets (Mirza & Osindero, 2014) are popular methods for this problem.

In this paper, we consider the problem of learning *all* conditional distributions of the form $p(x_I|x_{U \setminus I})$, where $U$ is the set of all features and $I$ is its arbitrary subset. This problem generalizes both learning the joint distribution $p(x)$ and learning the conditional distribution $p(x|y)$. To tackle this problem, we propose a *Variational Autoencoder with Arbitrary Conditioning* (VAEAC) model. It is a latent variable model similar to VAE, but allows conditioning on an arbitrary subset of the features. The conditioning features affect the prior on the latent Gaussian variables which are used to generate unobserved features. The model is trained using stochastic gradient variational Bayes (Kingma & Welling, 2013).

We consider two most natural applications of the proposed model. The first one is *feature imputation* where the goal is to restore the missing features given the observed ones. The imputed values may be valuable by themselves or may improve the performance of other machine learning algorithms which process the dataset. Another application is *image inpainting* in which the goal is to fill in an unobserved part of an image with an artificial content in a realistic way. This can be used for removing unnecessary objects from the images or, vice versa, for complementing the partially closed or corrupted object.

---

*Author is in DeepMind now.

The experimental evaluation shows that the proposed model successfully samples from the conditional distributions. The distribution over samples is close to the true conditional distribution. This property is very important when the true distribution has several modes. The model is shown to be effective in feature imputation problem which helps to increase the quality of subsequent discriminative models on different problems from UCI datasets collection (Lichman, 2013). We demonstrate that model can generate diverse and realistic image inpaintings on MNIST (LeCun et al., 1998), Omniglot (Lake et al., 2015) and CelebA (Liu et al., 2015) datasets, and works even better than the current state of the art inpainting techniques in terms of peak signal to noise ratio (PSNR).

The paper is organized as follows. In section 2 we review the related works. In section 3 we briefly describe variational autoencoders and conditional variational autoencoders. In section 4 we define the problem, describe the VAEAC model and its training procedure. In section 5 we evaluate VAEAC. Section 6 concludes the paper. Appendix contains additional explanations, theoretical analysis, and experiments for VAEAC.

## 2 RELATED WORK

Universal Marginalizer (Douglas et al., 2017) is a model based on a feed-forward neural network which approximates marginals of unobserved features conditioned on observable values. A related idea of an autoregressive model of joint probability was previously proposed in Germain et al. (2015) and Uria et al. (2016). The description of the model and comparison with VAEAC are available in section 5.3.

Yoon et al. (2018) propose a GANs-based model called GAIN which solves the same problem as VAEAC. In contrast to VAEAC, GAIN does not use unobserved data during training, which makes it easier to apply to the missing features imputation problem. Nevertheless, it turns into a disadvantage when the fully-observed training data is available but the missingness rate at the testing stage is high. For example, in inpainting setting GAIN cannot learn the conditional distribution over MNIST digits given one horizontal line of the image while VAEAC can (see appendix D.4). The comparison of VAEAC and GAIN on the missing feature imputation problem is given in section 5.1 and appendix D.2.

Rezende et al. (2014) [Appendix F], Sohl-Dickstein et al. (2015), Goyal et al. (2017), and Bordes et al. (2017) propose to fill missing data with noise and run Markov chain with a learned transition operator. The stationary distribution of such chains approximates the true conditional distribution of the unobserved features. Bachman & Precup (2015) consider missing feature imputation in terms of Markov decision process and propose LSTM-based sequential decision making model to solve it. Nevertheless, these methods are computationally expensive at the test time and require fully-observed training data.

Image inpainting is a classic computer vision problem. Most of the earlier methods rely on local and texture information or hand-crafted problem-specific features (Bertalmio et al., 2000). In past years multiple neural network based approaches have been proposed.

Pathak et al. (2016), Yeh et al. (2016) and Yang et al. (2017) use different kinds and combinations of adversarial, reconstruction, texture and other losses. Li et al. (2017) focuses on face inpainting and uses two adversarial losses and one semantic parsing loss to train the generative model. In Yeh et al. (2017) GANs are first trained on the whole training dataset. The inpainting is an optimization procedure that finds the latent variables that explain the observed features best. Then, the obtained latents are passed through the generative model to restore the unobserved portion of the image. We can say that VAEAC is a similar model which uses prior network to find a proper latents instead of solving the optimization problem.

All described methods aim to produce a single realistic inpainting, while VAEAC is capable of sampling diverse inpaintings. Additionally, Yeh et al. (2016), Yang et al. (2017) and Yeh et al. (2017) have high test-time computational complexity of inpainting, because they require an optimization problem to be solved. On the other hand, VAEAC is a "single-shot" method with a low computational cost.

## 3 BACKGROUND

### 3.1 VARIATIONAL AUTOENCODER

Variational autoencoder (Kingma & Welling, 2013) (VAE) is a directed generative model with latent variables. The generative process in variational autoencoder is as follows: first, a latent variable $z$ is generated from the *prior* distribution $p(z)$, and then the data $x$ is generated from the *generative* distribution $p_\theta(x|z)$, where $\theta$ are the generative model's parameters. This process induces the distribution $p_\theta(x) = \mathbb{E}_{p(z)} p_\theta(x|z)$. The distribution $p_\theta(x|z)$ is modeled by a neural network with parameters $\theta$. $p(z)$ is a standard Gaussian distribution.

The parameters $\theta$ are tuned by maximizing the likelihood of the training data points $\{x_i\}_{i=1}^N$ from the true data distribution $p_d(x)$. In general, this optimization problem is challenging due to intractable posterior inference. However, a variational lower bound can be optimized efficiently using backpropagation and stochastic gradient descent:

$$\log p_\theta(x) = \mathbb{E}_{q_\phi(z|x)} \log \frac{p_\theta(x,z)}{q_\phi(z|x)} + D_{\mathrm{KL}}(q_\phi(z|x)\|p(z|x,\theta))$$
$$\geq \mathbb{E}_{q_\phi(z|x)} \log p_\theta(x|z) - D_{\mathrm{KL}}(q_\phi(z|x)\|p(z)) = L_{VAE}(x;\theta,\phi) \quad (1)$$

Here $q_\phi(z|x)$ is a *proposal* distribution parameterized by neural network with parameters $\phi$ that approximates the posterior $p(z|x,\theta)$. Usually this distribution is Gaussian with a diagonal covariance matrix. The closer $q_\phi(z|x)$ to $p(z|x,\theta)$, the tighter variational lower bound $L_{VAE}(\theta,\phi)$. To compute the gradient of the variational lower bound with respect to $\phi$, reparameterization trick is used: $z = \mu_\phi(x) + \varepsilon\sigma_\phi(x)$ where $\varepsilon \sim \mathcal{N}(0, I)$ and $\mu_\phi$ and $\sigma_\phi$ are deterministic functions parameterized by neural networks. So the gradient can be estimated using Monte-Carlo method for the first term and computing the second term analytically:

$$\frac{\partial L_{VAE}(x;\theta,\phi)}{\partial\phi} = \mathbb{E}_{\varepsilon\sim\mathcal{N}(0,I)} \frac{\partial}{\partial\phi} \log p_\theta(x|\mu_\phi(x) + \varepsilon\sigma_\phi(x)) - \frac{\partial}{\partial\phi} D_{\mathrm{KL}}(q_\phi(z|x)\|p(z)) \quad (2)$$

So $L_{VAE}(\theta,\phi)$ can be optimized using stochastic gradient ascent with respect to $\phi$ and $\theta$.

### 3.2 CONDITIONAL VARIATIONAL AUTOENCODER

Conditional variational autoencoder (Sohn et al., 2015) (CVAE) approximates the conditional distribution $p_d(x|y)$. It outperforms deterministic models when the distribution $p_d(x|y)$ is multi-modal (diverse $x$s are probable for the given $y$). For example, assume that $x$ is a real-valued image. Then, a deterministic regression model with mean squared error loss would predict the average blurry value for $x$. On the other hand, CVAE learns the distribution of $x$, from which one can sample diverse and realistic objects.

Variational lower bound for CVAE can be derived similarly to VAE by conditioning all considered distributions on $y$:

$$L_{CVAE}(x,y;\theta,\psi,\phi) = \mathbb{E}_{q_\phi(z|x,y)} \log p_\theta(x|z,y) - D_{\mathrm{KL}}(q_\phi(z|x,y)\|p_\psi(z|y)) \leq \log p_{\theta,\psi}(x|y) \quad (3)$$

Similarly to VAE, this objective is optimized using the reparameterization trick. Note that the prior distribution $p_\psi(z|y)$ is conditioned on $y$ and is modeled by a neural network with parameters $\psi$. Thus, CVAE uses three trainable neural networks, while VAE only uses two.

Also authors propose such modifications of CVAE as Gaussian stochastic neural network and hybrid model. These modifications can be applied to our model as well. Nevertheless, we don't use them, because of their disadvantage which is described in appendix C.

## 4 VARIATIONAL AUTOENCODER WITH ARBITRARY CONDITIONING

### 4.1 PROBLEM STATEMENT

Consider a distribution $p_d(x)$ over a $D$-dimensional vector $x$ with real or categorical components. The components of the vector are called *features*.

Let binary vector $b \in \{0,1\}^D$ be the binary mask of *unobserved features* of the object. Then we describe the vector of unobserved features as $x_b = \{x_{i:b_i=1}\}$. For example, $x_{(0,1,1,0,1)} = (x_2, x_3, x_5)$. Using this notation we denote $x_{1-b}$ as a vector of *observed features*.

Our goal is to build a model of the conditional distribution $p_{\psi,\theta}(x_b|x_{1-b}, b) \approx p_d(x_b|x_{1-b}, b)$ for an arbitrary $b$, where $\psi$ and $\theta$ are parameters that are used in our model at the testing stage.

However, the true distribution $p_d(x_b|x_{1-b}, b)$ is intractable without strong assumptions about $p_d(x)$. Therefore, our model $p_{\psi,\theta}(x_b|x_{1-b}, b)$ has to be more precise for some $b$ and less precise for others. To formalize our requirements about the accuracy of our model we introduce the distribution $p(b)$ over different unobserved feature masks. The distribution $p(b)$ is arbitrary and may be defined by the user depending on the problem. Generally it should have full support over $\{0,1\}^D$ so that $p_{\psi,\theta}(x_b|x_{1-b}, b)$ can evaluate arbitrary conditioning. Nevertheless, it is not necessary if the model is used for specific kinds of conditioning (as we do in section 5.2).

Using $p(b)$ we can introduce the following log-likelihood objective function for the model:

$$\max_{\psi,\theta} \mathbb{E}_{p_d(x)} \mathbb{E}_{p(b)} \log p_{\psi,\theta}(x_b|x_{1-b}, b) \tag{4}$$

The special cases of the objective (4) are variational autoencoder ($b_i = 1 \; \forall i \in \{1, \ldots, D\}$) and conditional variational autoencoder ($b$ is constant).

### 4.2 MODEL DESCRIPTION

The generative process of our model is similar to the generative process of CVAE: for each object firstly we generate $z \sim p_\psi(z|x_{1-b}, b)$ using prior network, and then sample unobserved features $x_b \sim p_\theta(x_b|z, x_{1-b}, b)$ using generative network. This process induces the following model distribution over unobserved features:

$$p_{\psi,\theta}(x_b|x_{1-b}, b) = \mathbb{E}_{z \sim p_\psi(z|x_{1-b}, b)} p_\theta(x_b|z, x_{1-b}, b) \tag{5}$$

We use $z \in \mathbb{R}^d$, and Gaussian distribution $p_\psi$ over $z$, with parameters from a neural network with weights $\psi$: $p_\psi(z|x_{1-b}, b, \psi) = \mathcal{N}(z|\mu_\psi(x_{1-b}, b), \sigma_\psi^2(x_{1-b}, b)I)$. The real-valued components of distribution $p_\theta(x_b|z, x_{1-b}, b)$ are defined likewise. Each categorical component $i$ of distribution $p_\theta(x_i|z, x_{1-b}, b)$ is parameterized by a function $w_{i,\theta}(z, x_{1-b}, b)$, whose outputs are logits of probabilities for each category: $x_i \sim \text{Cat}[\text{Softmax}(w_{i,\theta}(z, x_{1-b}, b))]$. Therefore the components of the latent vector $z$ are conditionally independent given $x_{1-b}$ and $b$, and the components of $x_b$ are conditionally independent given $z$, $x_{1-b}$ and $b$.

The variables $x_b$ and $x_{1-b}$ have variable length that depends on $b$. So in order to use architectures such as multi-layer perceptron and convolutional neural network we consider $x_{1-b} = x \circ (1 - b)$ where $\circ$ is an element-wise product. So in implementation $x_{1-b}$ has fixed length. The output of the generative network also has a fixed length, but we use only unobserved components to compute likelihood.

The theoretical analysis of the model is available in appendix B.1.

### 4.3 Learning Variational Autoencoder with Arbitrary Conditioning

#### 4.3.1 Variational Lower Bound

We can derive a lower bound for $\log p_{\psi,\theta}(x_b|x_{1-b}, b)$ as for variational autoencoder:

$$
\log p_{\psi,\theta}(x_b|x_{1-b}, b) = \mathbb{E}_{q_\phi(z|x,b)} \log \frac{p_{\psi,\theta}(x_b, z|x_{1-b}, b)}{q_\phi(z|x, b)} + D_{\mathrm{KL}}(q_\phi(z|x, b)\|p_{\psi,\theta}(z|x, b))
$$

$$
\geq \mathbb{E}_{q_\phi(z|x,b)} \log p_\theta(x_b|z, x_{1-b}, b) - D_{\mathrm{KL}}(q_\phi(z|x, b)\|p_\psi(z|x_{1-b}, b)) = L_{VAEAC}(x, b; \theta, \psi, \phi) \quad (6)
$$

Therefore we have the following variational lower bound optimization problem:

$$
\max_{\theta,\psi,\phi} \mathbb{E}_{p_d(x)} \mathbb{E}_{p(b)} L_{VAEAC}(x, b; \theta, \psi, \phi) \quad (7)
$$

We use fully-factorized Gaussian proposal distribution $q_\phi$ which allows us to perform reparameterization trick and compute KL divergence analytically in order to optimize (7).

#### 4.3.2 Prior In Latent Space

During the optimization of objective (7) the parameters $\mu_\psi$ and $\sigma_\psi$ of the prior distribution of $z$ may tend to infinity, since there is no penalty for large values of those parameters. We usually observe the growth of $\|z\|_2$ during training, though it is slow enough. To prevent potential numerical instabilities, we put a Normal-Gamma prior on the parameters of the prior distribution to prevent the divergence. Formally, we redefine $p_\psi(z|x_{1-b}, b)$ as follows:

$$
p_\psi(z, \mu_\psi, \sigma_\psi|x_{1-b}, b) = \mathcal{N}(z|\mu_\psi, \sigma_\psi^2)\mathcal{N}(\mu_\psi|0, \sigma_\mu)\,\mathrm{Gamma}(\sigma_\psi|2, \sigma_\sigma) \quad (8)
$$

As a result, the regularizers $-\frac{\mu_\psi^2}{2\sigma_\mu^2}$ and $\sigma_\sigma(\log(\sigma_\psi) - \sigma_\psi)$ are added to the model log-likelihood. Hyperparameter $\sigma_\mu$ is chosen to be large ($10^4$) and $\sigma_\sigma$ is taken to be a small positive number ($10^{-4}$). This distribution is close to uniform near zero, so it doesn't affect the learning process significantly.

#### 4.3.3 Missing Features

The optimization objective (7) requires all features of each object at the training stage: some of the features will be observed variables at the input of the model and other will be unobserved features used to evaluate the model. Nevertheless, in some problem settings the training data contains missing features too. We propose the following slight modification of the problem (7) in order to cover such problems as well.

The missing values cannot be observed so $x_i = \omega \Rightarrow b_i = 1$, where $\omega$ describes the missing value in the data. In order to meet this requirement, we redefine mask distribution as conditioned on $x$: $p(b)$ turns into $p(b|x)$ in (4) and (7). In the reconstruction loss (5) we simply omit the missing features, i. e. marginalize them out:

$$
\log p_\theta(x_b|z, x_{1-b}, b) = \sum_{i:b_i=1, x_i \neq \omega} \log p_\theta(x_i|z, x_{1-b}, b) \quad (9)
$$

The proposal network must be able to determine which features came from real object and which are just missing. So we use additional missing features mask which is fed to proposal network together with unobserved features mask $b$ and object $x$.

The proposed modifications are evaluated in section 5.1.

Table 1: NRMSE (for continuous datasets) or PFC (for categorical ones) of imputations. Less is better.

| Method / Dataset | WhiteWine | Yeast | Mushroom | Zoo | Phishing |
|---|---|---|---|---|---|
| MICE | $0.964 \pm 0.007$ | $1.01 \pm 0.01$ | $0.334 \pm 0.002$ | $0.19 \pm 0.03$ | $0.422 \pm 0.006$ |
| MissForest | $0.878 \pm 0.009$ | $1.02 \pm 0.06$ | $0.249 \pm 0.006$ | $\mathbf{0.16 \pm 0.02}$ | $0.422 \pm 0.009$ |
| GAIN | $0.97 \pm 0.02$ | $0.99 \pm 0.03$ | $0.271 \pm 0.003$ | $0.20 \pm 0.02$ | $0.427 \pm 0.010$ |
| VAEAC | $\mathbf{0.850 \pm 0.007}$ | $\mathbf{0.94 \pm 0.01}$ | $\mathbf{0.244 \pm 0.002}$ | $\mathbf{0.16 \pm 0.02}$ | $\mathbf{0.394 \pm 0.006}$ |

## 5 EXPERIMENTS

In this section we validate the performance of VAEAC using several real-world datasets. In the first set of experiments we evaluate VAEAC missing features imputation performance using various UCI datasets (Lichman, 2013). We compare imputations from our model with imputations from such classical methods as MICE (Buuren & Groothuis-Oudshoorn, 2010) and MissForest (Stekhoven & Bühlmann, 2011) and recently proposed GANs-based method GAIN (Yoon et al., 2018). In the second set of experiments we use VAEAC to solve image inpainting problem. We show inpainitngs generated by VAEAC and compare our model with models from papers Pathak et al. (2016), Yeh et al. (2017) and Li et al. (2017) in terms of peak signal-to-noise ratio (PSNR) of obtained inpaintings on CelebA dataset (Liu et al., 2015) . And finally, we evaluate VAEAC against the competing method called Universal Marginalizer (Douglas et al., 2017). Additional experiments can be found in appendices C and D. The code is available at `https://github.com/tigvarts/vaeac`.

### 5.1 MISSING FEATURES IMPUTATION

The datasets with missing features are widespread. Consider a dataset with $D$-dimensional objects $x$ where each feature may be missing (which we denote by $x_i = \omega$) and their target values $y$. The majority of discriminative methods do not support missing values in the objects. The procedure of filling in the missing features values is called missing features imputation.

In this section we evaluate the quality of imputations produced by VAEAC. For evaluation we use datasets from UCI repository (Lichman, 2013). Before training we drop randomly 50% of values both in train and test set. After that we impute missing features using MICE (Buuren & Groothuis-Oudshoorn, 2010), MissForest (Stekhoven & Bühlmann, 2011), GAIN (Yoon et al., 2018) and VAEAC trained on the observed data. The details of GAIN implementation are described in appendix A.4.

Our model learns the distribution of the imputations, so it is able to sample from this distribution. We replace each object with missing features by $n = 10$ objects with sampled imputations, so the size of the dataset increases by $n$ times. This procedure is called missing features *multiple imputation*. MICE and GAIN are also capable of multiple imputation (we use $n = 10$ for them in experiments as well), but MissForest is not.

For more details about the experimental setup see appendices A.1, A.2, and A.4.

In table 1 we report NRMSE (i.e. RMSE normalized by the standard deviation of each feature and then averaged over all features) of imputations for continuous datasets and proportion of falsely classified (PFC) for categorical ones. For multiple imputation methods we average imputations of continuous variables and take most frequent imputation for categorical ones for each object.

We also learn linear or logistic regression and report the regression or classification performance after applying imputations of different methods in table 2. For multiple imputation methods we average predictions for continuous targets and take most frequent prediction for categorical ones for each object in test set.

Table 2: R2-score (for continuous targets) or accuracy (for categorical ones) of post-imputation regression or classification. Higher is better.

| Method / Dataset | WhiteWine | Yeast | Mushroom | Zoo | Phishing |
|---|---|---|---|---|---|
| MICE | $0.13 \pm 0.02$ | $\mathbf{0.41 \pm 0.02}$ | $0.92 \pm 0.01$ | $\mathbf{0.78 \pm 0.05}$ | $\mathbf{0.75 \pm 0.02}$ |
| MissForest | $\mathbf{0.17 \pm 0.01}$ | $\mathbf{0.42 \pm 0.02}$ | $0.972 \pm 0.003$ | $0.71 \pm 0.07$ | $0.73 \pm 0.02$ |
| GAIN | $0.11 \pm 0.01$ | $\mathbf{0.39 \pm 0.06}$ | $0.969 \pm 0.005$ | $0.67 \pm 0.06$ | $0.74 \pm 0.03$ |
| VAEAC | $\mathbf{0.17 \pm 0.01}$ | $\mathbf{0.43 \pm 0.01}$ | $\mathbf{0.983 \pm 0.002}$ | $\mathbf{0.8 \pm 0.1}$ | $0.74 \pm 0.02$ |

As can be seen from the tables 1 and 2, VAEAC can learn joint data distribution and use it for missing feature imputation. The imputations are competitive with current state of the art imputation methods in terms of RMSE, PFC, post-imputation regression R2-score and classification accuracy. Nevertheless, we don't claim that our method is state of the art in missing features imputation; for some datasets MICE or MissForest outperform it. The additional experiments can be found in appendix D.2.

## 5.2 IMAGE INPAINTING

The image inpainting problem has a number of different formulations. The formulation of our interest is as follows: some of the pixels of an image are unobserved and we want to restore them in a natural way. Unlike the majority of papers, we want to restore not just one most probable inpainting, but the distribution over all possible inpaintings from which we can sample. This distribution is extremely multi-modal because often there is a lot of different possible ways to inpaint the image.

Unlike the previous subsection, here we have uncorrupted images without missing features in the training set, so $p(b|x) = p(b)$.

As we show in section 2, state of the art results use different adversarial losses to achieve more sharp and realistic samples. VAEAC can be adapted to the image inpainting problem by using a combination of those adversarial losses as a part of reconstruction loss $p_\theta(x_b|z, x_{1-b}, b)$. Nevertheless, such construction is out of scope for this research, so we leave it for the future work. In the current work we show that the model can generate both diverse and realistic inpaintings.

In figures 1, 2, 3 and 4 we visualize image inpaintings produced by VAEAC on binarized MNIST (LeCun et al., 1998), Omniglot (Lake et al., 2015) and CelebA (Liu et al., 2015). The details of learning procedure and description of datasets are available in appendixes A.1 and A.3.

To the best of our knowledge, the most modern inpainting papers don't consider the *diverse* inpainting problem, where the goal is to build diverse image inpaintings, so there is no straightforward way to compare with these models. Nevertheless, we compute peak signal-to-noise ratio (PSNR) for one random inpainting from VAEAC and the best PSNR among 10 random inpaintings from VAEAC. One inpainting might not be similar to the original image, so we also measure how good the inpainting which is most similar to the original image reconstructs it. We compare these two metrics computed for certain masks with the PSNRs for the same masks on CelebA from papers Yeh et al. (2017) and Li et al. (2017). The results are available in tables 3 and 4.

We observe that for the majority of proposed masks our model outperforms the competing methods in terms of PSNR even with one sample, and for the rest (where the inpaintings are significantly diverse) the best PSNR over 10 inpaintings is larger than the same PSNR of the competing models. Even if PSNR does not reflect completely the visual quality of images and tends to encourage blurry VAE samples instead of realistic GANs samples, the results show that VAEAC is able to solve inpainting problem comparably to the state of the art methods. The disadvantage of VAEAC compared to Yeh et al. (2017) and Li et al. (2017) (but

Table 3: PSNR of inpaintings for different masks for Context Encoder (Pathak et al., 2016), model from "Semantic Image Inpainting with Deep Generative Models" (Yeh et al., 2017) and VAEAC. Higher is better.

| Method/Masks | Center | Pattern | Random | Half |
|---|---|---|---|---|
| Context Encoder [1] | 21.3 | 19.2 | 20.6 | 15.5 |
| SIIDGM [1] | 19.4 | 17.4 | 22.8 | 13.7 |
| VAEAC, 1 sample | 22.1 | 21.4 | 29.3 | 14.9 |
| VAEAC, 10 samples | 23.7 | 23.3 | 29.3 | 17.4 |

Table 4: PSNR of inpaintings for different masks for Context Encoder (Pathak et al., 2016), model from "Generative Face Completion" (Li et al., 2017) and VAEAC. Higher is better.

| Method/Masks | O1 | O2 | O3 | O4 | O5 | O6 |
|---|---|---|---|---|---|---|
| Context Encoder [2] | 18.6 | 18.4 | 17.9 | 19.0 | 19.1 | 19.3 |
| GFC [2] | 20.0 | 19.8 | 18.8 | 19.7 | 19.5 | 20.2 |
| VAEAC, 1 sample | 20.8 | 21.0 | 19.5 | 20.3 | 20.3 | 21.0 |
| VAEAC, 10 samples | 22.0 | 22.2 | 20.8 | 21.7 | 21.8 | 22.2 |

not Pathak et al. (2016)) is that it needs the distribution over masks at the training stage to be similar to the distribution over them at the test stage. However, it is not a very strict limitation for the practical usage.

## 5.3 Universal Marginalizer

Universal Marginalizer (Douglas et al., 2017) (UM) is a model which uses a single neural network to estimate the marginal distributions over the unobserved features. So it optimizes the following objective:

$$\max_{\theta} \mathbb{E}_{x \sim p_d(x)} \mathbb{E}_{b \sim p(b)} \sum_{i=1}^{D} b_i \log p_\theta(x_i | x_{1-b}, b) \tag{10}$$

For given mask $b$ we fix a permutation of its unobserved components: $(i_1, i_2, \ldots, i_{|b|})$, where $|b|$ is a number of unobserved components. Using the learned model and the permutation we can generate objects from joint distribution and estimate their probability using chain rule.

$$\log p_\theta(x_b | x_{1-b}, b) = \sum_{j=1}^{|b|} \log p_\theta(x_{i_j} | x_{1-(b-\sum_{k=1}^{j-1} e_{i_k})}, b - \sum_{k=1}^{j-1} e_{i_k}) \tag{11}$$

For example, $p_\theta(x_1, x_4, x_5 | x_2, x_3) = p_\theta(x_4 | x_2, x_3) p_\theta(x_1 | x_2, x_3, x_4) p_\theta(x_5 | x_1, x_2, x_3, x_4)$.

Conditional sampling or conditional likelihood estimation for one object requires $|b|$ requests to UM to compute $p_\theta(x_i | x_{1-b}, b)$. Each request is a forward pass through the neural network. In the case of conditional sampling those requests even cannot be paralleled because the input of the next request contains the output of the previous one.

We propose a slight modification of the original UM training procedure which allows learning UM efficiently for any kind of masks including those considered in this paper. The details of the modification are described in appendix B.3.

---

[1] The results are from the paper (Yeh et al., 2017)

[2] The results are from the paper (Li et al., 2017)

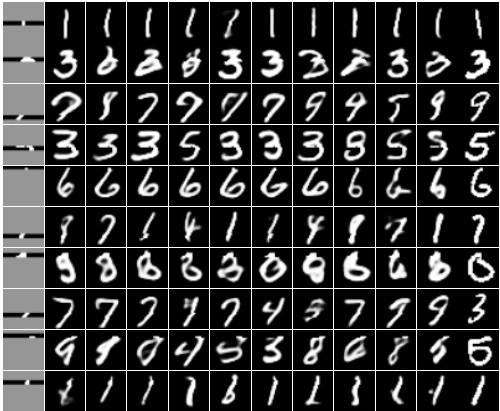

Figure 1: MNIST inpaintings.

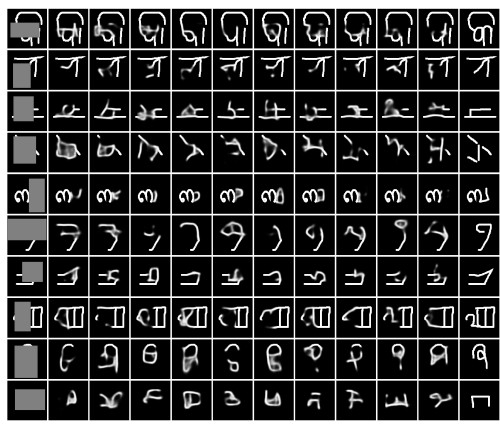

Figure 2: Omniglot inpaintings.

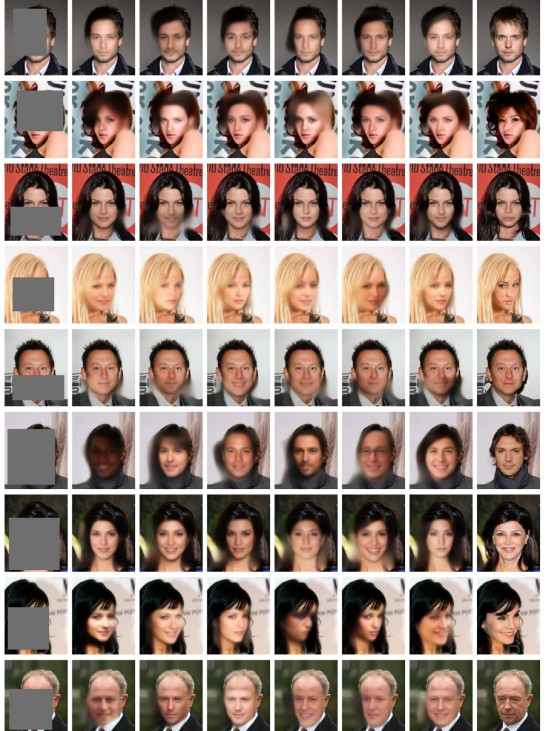

Figure 3: CelebA inpaintings.

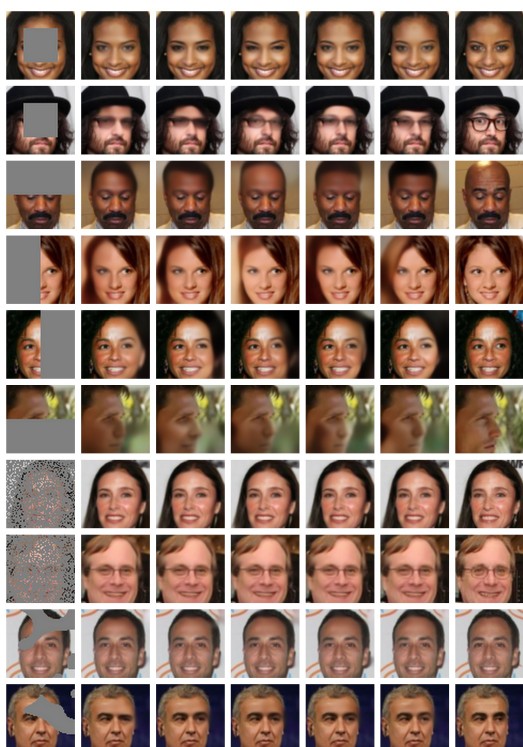

Figure 4: CelebA inpaintings with masks from (Yeh et al., 2017).

Left: input. The gray pixels are unobserved. Middle: samples from VAEAC. Right: ground truth.

Table 5: VAEAC and UM comparison on MNIST.

| Method | VAEAC | UM |
|---|---|---|
| Negative log-likelihood | 61 | 41 |
| Training time (30 epochs) | 5min 47s | 3min 14s |
| Test time (100 samples generation) | 0.7ms | 1s |

The results of using this modification of UM are provided in table 5. We can say that the relation between VAEAC and UM is similar to the relation between VAE and PixelCNN. The second one is much slower at the testing stage, but it easily takes into account local dependencies in data while the first one is faster but assumes conditional independence of the outputs. Nevertheless, there are a number of cases where UM cannot learn the distribution well while VAEAC can. For example, when the data is real-valued and marginal distributions have many local optima, there is no straightforward parametrization which allows UM to approximate them, and, therefore also the conditioned joint distribution. An example of such distribution and more illustrations for comparison of VAEAC and UM are available in appendix D.5.

## 6  CONCLUSION

In this paper we consider the problem of simultaneous learning of all conditional distributions for a vector. This problem has a number of different special cases with practical applications. We propose neural network based probabilistic model for distribution conditioning learning with Gaussian latent variables. This model is scalable and efficient in inference and learning. We propose several tricks to improve optimization and give recommendations about hyperparameters choice. The model is successfully applied to feature imputation and inpainting tasks. The experimental results show that the model is competitive with state of the art methods for both missing features imputation and image inpainting problems.

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

# APPENDIX

## A  EXPERIMENTAL DETAILS

### A.1  NEURAL NETWORK ARCHITECTURES

In all experiments we use optimization method Adam (Kingma & Ba, 2014), skip-connections between prior network and generative network inspired by (Mao et al., 2016), (Sønderby et al., 2016) and (Ronneberger et al., 2015), and convolutional neural networks based on ResNet blocks (He et al., 2016).

Without skip-connections all information for decoder goes through the latent variables. In image inpainting we found skip-connections very useful in both terms of log-likelihood improvement and the image realism, because latent variables are responsible for the global information only while the local information passes through skip-connections. Therefore the border between image and inpainting becomes less conspicuous.

The main idea of neural networks architecture is reflected in figure 5.

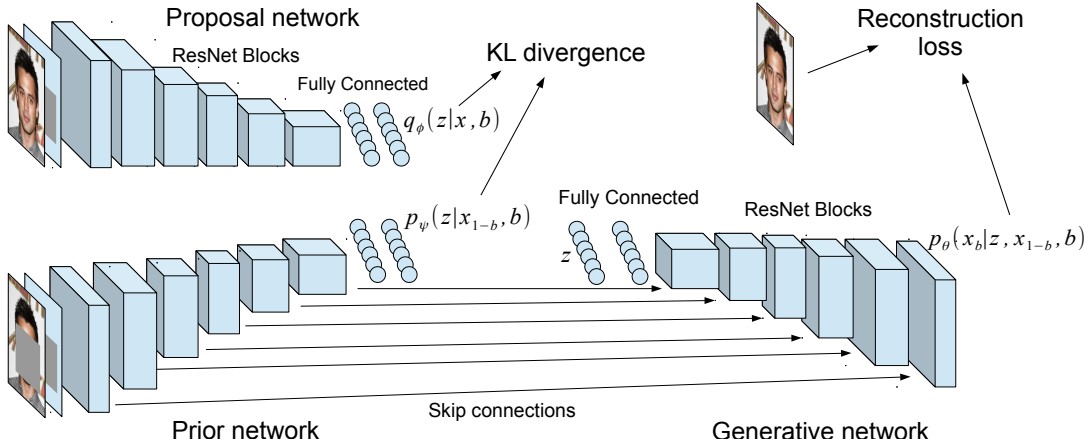

Figure 5: Neural network architecture for inpainting.

The number of hidden layers, their widths and structure may be different.

The neural networks we used for image inpainting have He-Uniform initialization of convolutional ResNet blocks, and the skip-connections are implemented using concatenation, not addition. The proposal network structure is exactly the same as the prior network except skip-connections.

Also one could use much simpler fully-connected networks with one hidden layer as a proposal, prior and generative networks in VAEAC and still obtain nice inpaintings on MNIST.

### A.2 MISSING FEATURES IMPUTATION

We split the dataset into train and test set with size ratio 3:1. Before training we drop randomly 50% of values both in train and test set. We repeat each experiment 5 times with different train-test splits and dropped features and then average results and compute their standard deviation.

As we show in appendix B.2, the better results can be achieved when the model learns the concatenation of objects features $x$ and targets $y$. So we treat $y$ as an additional feature that is always unobserved during the testing time.

To train our model we use distribution $p(b_i|x)$ in which $p(b_i|x_i = \omega) = 1$ and $p(b_i|x) = 0.2$ otherwise. Also for VAEAC trainig we normalize real-valued features, fix $\sigma_\theta = 1$ in the generative model of VAEAC in order to optimize RMSE, and use 25% of training data as validation set to select the best model among all epochs of training.

For the test set, the classifier or regressor is applied to each of the $n$ imputed objects and the predictions are combined. For regression problems we report R2-score of combined predictions, so we use averaging as a combination method. For classification problem we report accuracy, and therefore choose the mode. We consider the workflow where the imputed values of $y$ are not fed to the classifier or regressor to make a fair comparison of feature imputation quality.

Table 6: Generative Face Completion (Li et al., 2017) masks. Image size is 128x128.

| Mask | Meaning | $x_1$ | $x_2$ | $y_1$ | $y_2$ |
|------|---------|-------|-------|-------|-------|
| O1 | Left half of the face | 33 | 70 | 52 | 115 |
| O2 | Right half of the face | 57 | 70 | 95 | 115 |
| O3 | Two eyes | 29 | 98 | 52 | 73 |
| O4 | Left eye | 29 | 66 | 52 | 73 |
| O5 | Right eye | 61 | 99 | 52 | 73 |
| O6 | Lower half of the face | 40 | 87 | 86 | 123 |

NRMSE or PFC for dataset is computed as an average of NRMSE or PFC of all features of this dataset. NRMSE of a feature is just RMSE of imputations divided by the standard deviation of this feature. PFC of a feature is a proportion of imputations which are incorrect.

### A.3 IMAGE INPAINTING DATASETS AND MASKS

**MNIST** is a dataset of 60000 train and 10000 test grayscale images of digits from 0 to 9 of size 28x28. We binarize all images in the dataset. For MNIST we consider Bernoulli log-likelihood as the reconstruction loss: $\log p_\theta(x_b|z, x_{1-b}, b) = \sum_{i:b_i=1} \log \text{Bernoulli}(x_i|p_{\theta,i}(z, x_{1-b}, b))$ where $p_{\theta,i}(z, x_{1-b}, b)$ is an output of the generative neural network. We use 16 latent variables. In the mask for this dataset the observed pixels form a three pixels wide horizontal line which position is distributed uniformly.

**Omniglot** is a dataset of 19280 train and 13180 test black-and-white images of different alphabets symbols of size 105x105. As in previous section, the brightness of each pixel is treated as a Bernoulli probability of it to be 1. The mask we use is a random rectangular which is described below. We use 64 latent variables. We train model for 50 epochs and choose best model according to IWAE log-likelihood estimation on the validation set after each epoch.

**CelebA** is a dataset of 162770 train, 19867 validation and 19962 test color images of faces of celebrities of size 178x218. Before learning we normalize the channels in dataset. We use logarithm of fully-factorized Gaussian distribution as reconstruction loss. The mask we use is a random rectangular which is describe below. We use 32 latent variables.

**Rectangular mask** is the common shape of unobserved region in image inpainting. We use such mask for Omniglot and Celeba. We sample the corner points of rectangles uniprobably on the image, but reject those rectangles which area is less than a quarter of the image area.

**In Li et al. (2017)** six different masks O1–O6 are used on the testing stage. We reconstruct the positions of masks from the illustrations in the paper and give their coordinates in table 6. The visualizations of the masks are available in figure 10.

At the training stage we used a rectangle mask with uniprobable random corners. We reject masks with width or height less than 16pt. We use 64 latent variables and take the best model over 50 epochs based on the validation IWAE log-likelihood estimation. We can obtain slightly higher PSNR values than reported in table 4 if use only masks O1–O6 at the training stage.

**In Yeh et al. (2017)** four types of masks are used. Center mask is just an unobserved 32x32 square in the center of 64x64 image. Half mask mean that one of upper, lower, left or right half of the image is unobserved. All these types of a half are equiprobable. Random mask means that we use pixelwise-independent Bernoulli

distribution with probability 0.8 to form a mask of unobserved pixels. Pattern mask is proposed in Pathak et al. (2016). As we deduced from the code [3], the generation process is follows: firstly we generate 600x600 one-channel image with uniform distribution over pixels, then bicubically interpolate it to image of size 10000x10000, and then apply Heaviside step function $H(x - 0.25)$ (i. e. all points with value less than 0.25 are considered as unobserved). To sample a mask we sample a random position in this 10000x10000 binary image and crop 64x64 mask. If less than 20% or more than 30% of pixel are unobserved, than the mask is rejected and the position is sampled again. In comparison with this paper in section 5.2 we use the same distribution over masks at training and testing stages. We use VAEAC with 64 latent variables and take the best model over 50 epochs based on the validation IWAE log-likelihood estimation.

## A.4 GAIN IMPLEMENTATION DETAILS

For missing feature imputation we reimplemented GAIN in PyTorch based on the paper (Yoon et al., 2018) and the available TensorFlow source code for image inpainting [4].

For categorical features we use one-hot encoding. We observe in experiments that it works better in terms of NRMSE and PFC than processing categorical features in GAIN as continuous ones and then rounding them to the nearest category.

For categorical features we also use reconstruction loss $L_M(x_i, x_i') = -\frac{1}{|X_i|} \sum_{j=1}^{|X_i|} x_{i,j} \log(x_{i,j}')$. $|X_i|$ is the number of categories of the $i$-th feature, and $x_{i,j}$ is the $j$-th component of one-hot encoding of the feature $x_i$. Such $L_M$ enforces equal contribution of each categorical feature into the whole reconstruction loss.

We use one more modification of $L_M(x, x')$ for binary and categorical features. Cross-entropy loss in $L_M$ penalizes incorrect reconstructions of categorical and binary features much more than incorrect reconstructions for continuous ones. To avoid such imbalance we mixed L2 and cross-entropy reconstruction losses for binary and categorical features with weights 0.8 and 0.2 respectively:

$$L_M'(x_i, x_i') = 0.2 \cdot L_M(x_i, x_i') + 0.8 \cdot \begin{cases} \frac{1}{|X_i|} \sum_{j=1}^{|X_i|} (x_{i,j} - x_{i,j}')^2, \text{if } x_i \text{ is categorical} \\ (x_i - x_i')^2, \text{if } x_i \text{ is binary} \end{cases} \quad (12)$$

We observe in experiments that this modification also works better in terms of NRMSE and PFC than the original model.

We use validation set which contains 5% of the observed features for the best model selection (hyper-parameter is the number of iterations).

In the original GAIN paper authors propose to use cross-validation for hyper-parameter $\alpha \in \{0.1, 0.5, 1, 2, 10\}$. We observe that using $\alpha = 10$ and a hint $h = b \circ m + 0.5(1 - b)$ where vector $b$ is sampled from Bernoulli distribution with $p = 0.01$ provides better results in terms of NRMSE and PFC than the original model with every $\alpha \in \{0.1, 0.5, 1, 2, 10\}$. Such hint distribution makes model theoretically inconsistent but works well in practice (see table 7).

Table 7 shows that our modifications provide consistently not worse or even better imputations than the original GAIN (in terms of NRMSE and PFC, on the considered datasets). So in this paper for the missing feature imputation problem we report the results of our modification of GAIN.

---

[3] https://github.com/pathak22/context-encoder/blob/master/train_random.lua#L273

[4] https://github.com/jsyoon0823/GAIN

Table 7: NRMSE (for continuous datasets) or PFC (for categorical ones) of imputations for different GAIN modifications. Less is better. "Our modification" includes the reconstruction loss $L'_M$ (12), Bernoulli distribution over $b$ in the hint generation procedure, and fixed $\alpha = 10$. Other columns refers original GAIN without these modifications and with different values of $\alpha$.

| Dataset | Our modification | $\alpha = 10$ | $\alpha = 2$ | $\alpha = 1$ | $\alpha = 0.5$ | $\alpha = 0.1$ |
|---|---|---|---|---|---|---|
| Boston | $\mathbf{0.78 \pm 0.03}$ | $0.87 \pm 0.02$ | $1.0 \pm 0.1$ | $1.0 \pm 0.1$ | $1.02 \pm 0.05$ | $1.6 \pm 0.2$ |
| Breast | $\mathbf{0.67 \pm 0.01}$ | $0.80 \pm 0.05$ | $1.00 \pm 0.05$ | $1.10 \pm 0.07$ | $1.19 \pm 0.05$ | $1.52 \pm 0.06$ |
| Concrete | $\mathbf{0.96 \pm 0.01}$ | $\mathbf{0.98 \pm 0.02}$ | $1.02 \pm 0.02$ | $1.13 \pm 0.06$ | $1.17 \pm 0.04$ | $1.3 \pm 0.1$ |
| Diabetes | $\mathbf{0.911 \pm 0.009}$ | $\mathbf{0.93 \pm 0.03}$ | $1.05 \pm 0.04$ | $1.07 \pm 0.07$ | $1.21 \pm 0.07$ | $1.6 \pm 0.1$ |
| Digits | $\mathbf{0.79 \pm 0.02}$ | $0.88 \pm 0.01$ | $1.05 \pm 0.02$ | $1.13 \pm 0.02$ | $1.24 \pm 0.08$ | $1.4 \pm 0.2$ |
| Glass | $\mathbf{1.06 \pm 0.05}$ | $\mathbf{1.04 \pm 0.05}$ | $1.19 \pm 0.06$ | $1.4 \pm 0.2$ | $1.6 \pm 0.1$ | $1.81 \pm 0.10$ |
| Iris | $\mathbf{0.72 \pm 0.04}$ | $\mathbf{0.73 \pm 0.06}$ | $0.83 \pm 0.08$ | $0.97 \pm 0.09$ | $1.2 \pm 0.2$ | $1.3 \pm 0.2$ |
| Mushroom | $\mathbf{0.271 \pm 0.003}$ | $0.404 \pm 0.004$ | $0.52 \pm 0.05$ | $0.55 \pm 0.01$ | $0.56 \pm 0.03$ | $0.64 \pm 0.06$ |
| Orthopedic | $\mathbf{0.91 \pm 0.03}$ | $\mathbf{0.91 \pm 0.08}$ | $1.1 \pm 0.1$ | $1.2 \pm 0.1$ | $1.34 \pm 0.08$ | $1.6 \pm 0.2$ |
| Phishing | $\mathbf{0.427 \pm 0.010}$ | $0.52 \pm 0.02$ | $0.54 \pm 0.02$ | $0.543 \pm 0.010$ | $0.56 \pm 0.01$ | $0.57 \pm 0.04$ |
| WallRobot | $\mathbf{0.907 \pm 0.005}$ | $0.924 \pm 0.005$ | $0.933 \pm 0.008$ | $0.95 \pm 0.01$ | $1.00 \pm 0.02$ | $1.26 \pm 0.04$ |
| WhiteWine | $\mathbf{0.97 \pm 0.02}$ | $1.02 \pm 0.04$ | $1.2 \pm 0.1$ | $1.3 \pm 0.1$ | $1.6 \pm 0.1$ | $1.86 \pm 0.08$ |
| Yeast | $\mathbf{0.99 \pm 0.03}$ | $1.3 \pm 0.2$ | $1.6 \pm 0.1$ | $1.83 \pm 0.09$ | $1.9 \pm 0.1$ | $2.4 \pm 0.4$ |
| Zoo | $\mathbf{0.20 \pm 0.02}$ | $\mathbf{0.24 \pm 0.05}$ | $0.35 \pm 0.06$ | $0.36 \pm 0.03$ | $0.43 \pm 0.04$ | $0.433 \pm 0.004$ |

## B  THEORY

### B.1  VAEAC UNIVERSALITY

The theoretical guarantees that VAEAC can model arbitrary distribution are based on the same guarantees for Condtitional Variational Autoencoder (CVAE). We prove below that if CVAE can model each of the conditional distributions $p(x_b|x_{1-b})$, then VAEAC can model all of them.

We can imagine $2^D$ CVAEs learned each for the certain mask. Because neural networks are universal approximators, VAEAC networks could model the union of CVAE networks, so that VAEAC network performs transformation defined by the same network of the corresponding to the given mask CVAE.

$$p_{\psi,VAEAC}(z|x_{1-b}, b) = p_{\psi,CVAE,1-b}(z|x_{1-b}) \ \forall x, b$$

$$p_{\theta,VAEAC}(x_b|z, x_{1-b}, b) = p_{\theta,CVAE,1-b}(x_b|z, x_{1-b}) \ \forall z, x, b$$

So if CVAE models any distribution $p(x|y)$, VAEAC also do.

The guarantees for CVAE in the case of continuous variables are based on the point that every smooth distribution can be approximated with a large enough mixture of Gaussians, which is a special case of CVAE's generative model. These guarantees can be extended on the case of categorical-continuous variables also. Actually, there are distributions over categorical variables which CVAE with Gaussian prior and proposal distributions cannot learn. Nevertheless, this kind of limitation is not fundamental and is caused by poor proposal distribution family.

### B.2  WHY VAEAC NEEDS TARGET VALUES FOR MISSING FEATURES IMPUTATION?

Consider a dataset with $D$-dimensional objects $x$ where each feature may be missing (which we denote by $x_i = \omega$) and their target values $y$. In this section we show that the better results are achieved when our model learns the concatenation of objects features $x$ and targets $y$. The example that shows the necessity of it is following. Consider a dataset where $x_1 = 1$, $x_2 \sim \mathcal{N}(x_2|y, 1)$, $p_d(y = 0) = p(y = 5) = 0.5$. In this

case $p_d(x_2|x_1 = 1) = 0.5\mathcal{N}(x_2|0,1) + 0.5\mathcal{N}(x_2|5,1)$. We can see that generating data from $p_d(x_2|x_1)$ may only confuse the classifier, because with probability 0.5 it generates $x_2 \sim \mathcal{N}(0,1)$ for $y = 5$ and $x_2 \sim \mathcal{N}(5,1)$ for $y = 0$. On the other hand, $p_d(x_2|x_1, y) = \mathcal{N}(x_2|y, 1)$. Filling gaps using $p_d(x_2|x_1, y)$ may only improve classifier or regressor by giving it some information from the joint distribution $p_d(x, y)$ and thus simplifying the dependence to be learned at the training time. So we treat $y$ as an additional feature that is always unobserved during the testing time.

### B.3 Universal Marginalizer: Training Procedure Modification

The problem authors did not address in the original paper is the relation between the distribution of unobserved components $p(b)$ at the testing stage and the distribution of masks in the requests to UM $\hat{p}(b)$. The distribution over masks $p(b)$ induces the distribution $\hat{p}(b)$, and in the most cases $p(b) \neq \hat{p}(b)$. The distribution $\hat{p}(b)$ also depends on the permutations $(i_1, i_2, \ldots, i_{|b|})$ that we use to generate objects.

We observed in experiments, that UM must be trained using unobserved mask distribution $\hat{p}(b)$. For example, if all masks from $p(b)$ have a fixed number of unobserved components (e. g., $\frac{D}{2}$), then UM will never see an example of mask with $1, 2, \ldots, \frac{D}{2} - 1$ unobserved components, which is necessary to generate a sample conditioned on $\frac{D}{2}$ components. That leads to drastically low likelihood estimate for the test set and unrealistic samples.

We developed an easy generative process for $\hat{p}(b)$ for arbitrary $p(b)$ if the permutation of unobserved components $(i_1, i_2, \ldots, i_{|b|})$ is chosen randomly and equiprobably: firstly we generate $b_0 \sim p(b)$, $u \sim U[0,1]$, then $b_1 \sim (\text{Bernoulli}(u))^D$ and $b = b_0 \circ b_1$. More complicated generative process exists for a sorted permutation where $i_{j-1} < i_j \ \forall j : 2 \leq j \leq |b|$.

In experiments we use uniform distribution over the permutations.

## C  Gaussian Stochastic Neural Network

Gaussian stochastic neural network (13) and hybrid model (14) are originally proposed in the paper on Conditional VAE (Sohn et al., 2015). The motivation authors mention in the paper is as follows. During training the proposal distribution $q_\phi(z|x, y)$ is used to generate the latent variables $z$, while during the testing stage the prior $p_\psi(z|y)$ is used. KL divergence tries to close the gap between two distributions but, according to authors, it is not enough. To overcome the issue authors propose to use a *hybrid* model (14), a weighted mixture of variational lower bound (3) and a single-sample Monte-Carlo estimation of log-likelihood (13). The model corresponding to the second term is called Gaussian Stochastic Neural Network (13), because it is a feed-forward neural network with a single Gaussian stochastic layer in the middle. Also GSNN is a special case of CVAE where $q_\phi(z|x, y) = p_\psi(z|y)$.

$$L_{GSNN}(x, y; \theta, \psi) = \mathbb{E}_{p_\psi(z|y)} \log p_\theta(x|z, y) \tag{13}$$

$$L(x, y; \theta, \psi, \phi) = \alpha L_{CVAE}(x, y; \theta, \psi, \phi) + (1 - \alpha)L_{GSNN}(x, y; \theta, \psi), \quad \alpha \in [0, 1] \tag{14}$$

Authors report that hybrid model and GSNN outperform CVAE in terms of segmentation accuracy on the majority of datasets.

We can also add that this technique seems to soften the "holes problem" (Makhzani et al., 2016). In Makhzani et al. (2016) authors observe that vectors $z$ from prior distribution may be different enough from all vectors $z$ from the proposal distribution at the training stage, so the generator network may be confused at the testing stage. Due to this problem CVAE can have good reconstructions of $y$ given $z \sim q_\phi(z|x, y)$, while samples of $y$ given $z \sim p_\psi(z|x)$ are not realistic.

The same trick is applicable to our model as well:

$$L_{GSNN}(x, b; \theta, \psi) = \mathbb{E}_{p_\psi(z|x_{1-b}, b)} \log p_\theta(x_b | z, x_{1-b}, b) \tag{15}$$

$$L(x, b; \theta, \psi, \phi) = \alpha L_{VAEAC}(x, b; \theta, \psi, \phi) + (1 - \alpha) L_{GSNN}(x, b; \theta, \psi), \quad \alpha \in [0, 1] \tag{16}$$

In order to reflect the difference between sampling $z$ from prior and proposal distributions, authors of CVAE use two methods of log-likelihood estimation:

$$\log p_{\theta, \psi}(x|y) \approx \log \frac{1}{S} \sum_{i=1}^{S} p_\theta(x|z_i, y), \quad z_i \sim p_\psi(z|y) \tag{17}$$

$$\log p_{\theta, \psi}(x|y) \approx \log \frac{1}{S} \sum_{i=1}^{S} \frac{p_\theta(x|z_i, y) p_\psi(z_i|y)}{q_\phi(z_i|x, y)}, \quad z_i \sim q_\phi(z|x, y) \tag{18}$$

The first estimator is called *Monte-Carlo* estimator and the second one is called *Importance Sampling* estimator (also known as IWAE). They are asymptotically equivalent, but in practice the Monte-Carlo estimator requires much more samples to obtain the same accuracy of estimation. Small $S$ leads to underestimation of the log-likelihood for both Monte-Carlo and Importance Sampling (Burda et al., 2015), but for Monte-Carlo the underestimation is expressed much stronger.

We perform an additional study of GSNN and hybrid model and show that they have drawbacks when the target distribution $p(x|y)$ is has multiple different local maximums.

## C.1   THEORETICAL STUDY

In this section we show why GSNN cannot learn distributions with several different modes and leads to a blurry image samples.

For the simplicity of the notation we consider hybrid model for a standard VAE:

$$L(x; \phi, \psi, \theta) = \alpha \mathbb{E}_{z \sim q_\phi(z|x)} \log \frac{p_\theta(x|z) p_\psi(z)}{q_\phi(z|x)} + (1 - \alpha) \mathbb{E}_{z \sim p_\psi(z)} \log p_\theta(x|z) \tag{19}$$

The hybrid model (16) for VAEAC can be obtained from (19) by replacing $x$ with $x_b$ and conditioning all distributions on $x_{1-b}$ and $b$. The validity of the further equations and conclusions remains for VAEAC after this replacement.

Consider now a categorical latent variable $z$ which can take one of $K$ values. Let $x$ be a random variable with true distribution $p_d(x)$ to be modeled. Consider the following true data distribution: $p_d(x = x_i) = \frac{1}{K}$ for $i \in \{1, 2, \ldots, K\}$ and some values $x_1, x_2, \ldots, x_K$. So the true distribution has $K$ different equiprobable modes. Suppose the generator network $NN_\theta$ which models mapping from $z$ to some vector of parameters $v_z = NN_\theta(z)$. Thus, we define generative distribution as some function of these parameters: $p_\theta(x|z) = f(x, v_z)$. Therefore, the parameters $\theta$ are just the set of $v_1, v_2, \ldots, v_K$.

For the simplicity of the model we assume $p_\psi(z) = \frac{1}{K}$. Taking into account $p_\psi(z) = \frac{1}{K}$, we obtain optimal $q(z = i|x) = \frac{f(x, v_i)}{\sum_{j=1}^{K} f(x, v_j)}$. Using (19) and the above formulas for $q_\phi$, $p_\psi$ and $p_\theta$ we obtain the following optimization problem:

$$\max_{v_1, v_2, \ldots, v_K} \frac{1}{K} \sum_{i=1}^{K} \left[ \alpha \sum_{j=1}^{K} \frac{f(x_i, v_j)}{\sum_{k=1}^{K} f(x_i, v_k)} \log \frac{f(x_i, v_j) \frac{1}{K}}{\frac{f(x_i, v_j)}{\sum_{k=1}^{K} f(x_i, v_k)}} + (1 - \alpha) \sum_{j=1}^{K} \frac{1}{K} \log f(x_i, v_j) \right] \tag{20}$$

Table 8: Negative log-likelihood estimation of a hybrid model on the synthetic data. IS-$S$ refers to Importance Sampling log-likelihood estimation with $S$ samples for each object (18). MC-$S$ refers to Monte-Carlo log-likelihood estimation with $S$ samples for each object (17).

| VAEAC weight | IS-10 | MC-10 |
|---|---|---|
| $\alpha = 1$ | 0.22 | 85 |
| $\alpha = 0.99$ | 0.35 | 11 |
| $\alpha = 0.9$ | 0.62 | 1.7 |

It is easy to show that (20) is equivalent to the following optimization problem:

$$\max_{v_1, v_2, \ldots, v_K} \sum_{i=1}^{K} \left[ \alpha \log \frac{\sum_{j=1}^{K} f(x_i, v_j)}{K} + (1 - \alpha) \sum_{j=1}^{K} \frac{1}{K} \log f(x_i, v_j) \right] \qquad (21)$$

It is clear from (21) that when $\alpha = 1$ the log-likelihood of the initial model is optimized. On the other hand, when $\alpha = 0$ the optimal point is $v_1 = v_2 = \cdots = v_K = \arg\max_v \sum_{i=1}^{K} \log f(x_i, v)$, i. e. $z$ doesn't influence the generative process, and for each $z$ generator produces the same $v$ which maximizes likelihood estimation of the generative model $f(x, v)$ for the given dataset of $x$'s. For Bernoulli and Gaussian generative distributions $f$ such $v$ is just average of all modes $x_1, x_2, \ldots, x_K$. That explains why further we observe blurry images when using GSNN model.

The same conclusion holds for for continuous latent variables instead of categorical. Given $K$ different modes in true data distribution, VAE uses proposal network to separate prior distribution into $K$ components (i. e. regions in the latent space), so that each region corresponds to one mode. On the other hand, in GSNN $z$ is sampled independently on the mode which is to be reconstructed from it, so for each $z$ the generator have to produce parameters suitable for all modes.

From this point of view, there is no difference between VAE and VAEAC. If the true conditional distribution has several different modes, then VAEAC can fit them all, while GSNN learns their average. If true conditional distribution has one mode, GSNN and VAEAC are equal, and GSNN may even learn faster because it has less parameters.

Hybrid model is a trade-off between VAEAC and GSNN: the closer $\alpha$ to zero, the more blurry and closer to the average is the distribution of the model. The exact dependence of the model distribution on $\alpha$ can be derived analytically for the simple data distributions or evaluated experimentally. We perform such experimental evaluation in the next sections.

## C.2  SYNTHETIC DATA

In this section we show that VAEAC is capable of learning a complex multimodal distribution of synthetic data while GSNN and hybrid model are not. Let $x \in \mathbb{R}^2$ and $p(b_1 = 1) = p(b_2 = 1) = 0.5$. $p_d(x) = \frac{1}{8} \sum_{i=1}^{8} \mathcal{N}(x|\mu_i, \frac{1}{10}I)$ where $\mu_i \sim \mathcal{N}(\mu_i|0, I)$. The distribution $p(x)$ is plotted in figure 6. The dataset contains 100000 points sampled from $p_d(x)$. We use multi-layer perceptron with four ReLU layers of size 400-200-100-50, 25-dimensional Gaussian latent variables.

For different mixture coefficients $\alpha$ we visualize samples from the learned distributions $p_{\psi,\theta}(x_1, x_2)$, $p_{\psi,\theta}(x_1|x_2)$, and $p_{\psi,\theta}(x_2|x_1)$. The observed features for the conditional distributions are generated from the marginal distributions $p(x_2)$ and $p(x_1)$ respectively.

We see in table 8 and in figure 7, that even with very small weight GSNN prevents model from learning distributions with several local optimas. GSNN also increases Monte-Carlo log-likelihood estimation with

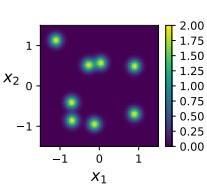

Figure 6: Probability density function of synthetic data distribution.

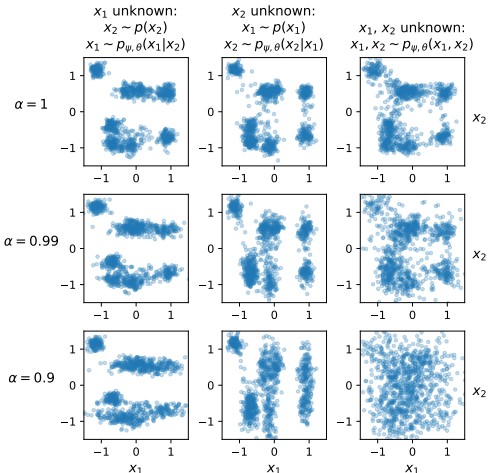

Figure 7: VAEAC for synthetic data.

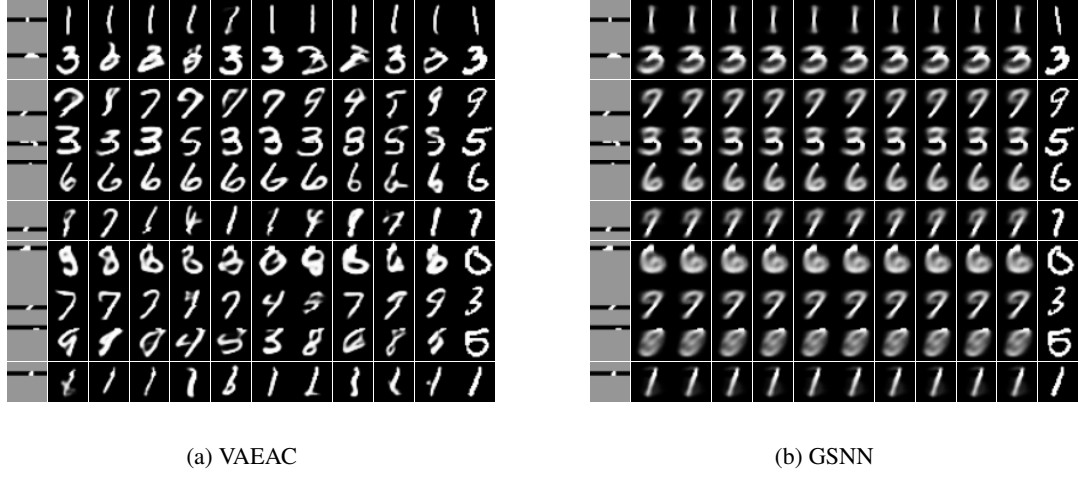

(a) VAEAC                                (b) GSNN

Figure 8: MNIST inpaintings.
Left: input. The gray pixels are unobserved. Middle: samples from the model. Right: ground truth.

a few samples and decreases much more precise Importance Sampling log-likelihood estimation. When $\alpha = 0.9$ the whole distribution structure is lost.

We see that using $\alpha \neq 1$ ruins multimodality of the restored distribution, so we highly recommend to use $\alpha = 1$ or at least $\alpha \approx 1$.

Table 9: Average negative log-likelihood of inpaintings for 1000 objects. IS-$S$ refers to Importance Sampling log-likelihood estimation with $S$ samples for each object (18). MC-$S$ refers to Monte-Carlo log-likelihood estimation with $S$ samples for each object (17). Naive Bayes is a baseline method which assumes pixels and colors independence.

| Method | MNIST | Omniglot | CelebA |
|---|---|---|---|
| VAEAC IS-$10^2$ | $\mathbf{61} \pm 1$ | $\mathbf{275} \pm 17$ | $\mathbf{34035} \pm 1609$ |
| VAEAC MC-$10^4$ | $94 \pm 4$ | $1452 \pm 109$ | $41513 \pm 2163$ |
| VAEAC MC-$10^2$ | $156 \pm 1$ | $2203 \pm 150$ | $53904 \pm 3121$ |
| GSNN MC-$10^4$ | $141 \pm 7$ | $1199 \pm 62$ | $53427 \pm 2208$ |
| GSNN MC-$10^2$ | $141 \pm 1$ | $1200 \pm 62$ | $53486 \pm 2210$ |
| Naive Bayes | $205$ | $2490$ | $269480$ |

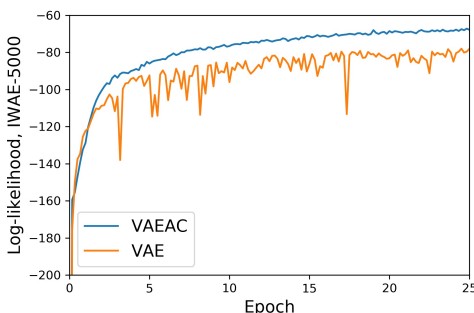

Figure 9: Convergence of VAE and VAEAC on MNIST dataset.

## C.3    COMPARISON ON THE IMAGE INPAINTING PROBLEM

In figure 8 we can see that the inpaintings produced by GSNN are smooth, blurry and not diverse compared with VAEAC.

Table 9 shows that VAEAC learns distribution over inpaintings better than GSNN in terms of test log-likelihood. Nevertheless, Monte-Carlo estimations with a small number of samples sometimes are better for GSNN, which means less local modes in the learned distribution and more blurriness in the samples.

## D    ADDITIONAL EXPERIMENTS

### D.1    CONVERGENCE SPEED

In figure 9 one can see that VAEAC has similar convergence speed to VAE in terms of iterations on MNIST dataset. In our experiments we observed the same behaviour for other datasets. Each iteration of VAEAC is about 1.5 times slower than VAE due to usage of three networks instead of two.

Table 10: NRMSE (for continuous datasets) or PFC (for categorical ones) of imputations. Less is better.

| Dataset | MICE | MissForest | GAIN | VAEAC | GSNN | NN |
|---------|------|------------|------|-------|------|-----|
| Boston | $0.69 \pm 0.02$ | $\mathbf{0.58 \pm 0.02}$ | $0.78 \pm 0.03$ | $0.71 \pm 0.02$ | $0.70 \pm 0.01$ | $0.69 \pm 0.01$ |
| Breast | $0.58 \pm 0.02$ | $\mathbf{0.515 \pm 0.008}$ | $0.67 \pm 0.01$ | $0.55 \pm 0.02$ | $0.55 \pm 0.02$ | $\mathbf{0.52 \pm 0.02}$ |
| Concrete | $0.850 \pm 0.007$ | $\mathbf{0.78 \pm 0.01}$ | $0.96 \pm 0.01$ | $0.84 \pm 0.02$ | $0.85 \pm 0.01$ | $\mathbf{2 \pm 3}$ |
| Diabetes | $\mathbf{0.80 \pm 0.01}$ | $0.84 \pm 0.02$ | $0.911 \pm 0.009$ | $0.90 \pm 0.03$ | $0.91 \pm 0.03$ | $0.90 \pm 0.02$ |
| Digits | $0.69 \pm 0.02$ | $\mathbf{0.61 \pm 0.02}$ | $0.79 \pm 0.02$ | $0.69 \pm 0.02$ | $0.69 \pm 0.02$ | $0.67 \pm 0.02$ |
| Glass | $0.91 \pm 0.02$ | $\mathbf{0.83 \pm 0.04}$ | $1.06 \pm 0.05$ | $0.91 \pm 0.04$ | $0.91 \pm 0.05$ | $\mathbf{0.87 \pm 0.04}$ |
| Iris | $\mathbf{0.59 \pm 0.02}$ | $0.62 \pm 0.04$ | $0.72 \pm 0.04$ | $0.64 \pm 0.04$ | $\mathbf{0.62 \pm 0.04}$ | $0.61 \pm 0.02$ |
| Mushroom | $0.334 \pm 0.002$ | $0.249 \pm 0.006$ | $0.271 \pm 0.003$ | $\mathbf{0.241 \pm 0.002}$ | $0.2412 \pm 0.0009$ | $\mathbf{0.239 \pm 0.001}$ |
| Orthopedic | $\mathbf{0.76 \pm 0.02}$ | $\mathbf{0.79 \pm 0.03}$ | $0.91 \pm 0.03$ | $0.80 \pm 0.03$ | $0.81 \pm 0.03$ | $0.81 \pm 0.02$ |
| Phishing | $0.422 \pm 0.006$ | $0.422 \pm 0.009$ | $0.427 \pm 0.010$ | $\mathbf{0.397 \pm 0.010}$ | $\mathbf{0.392 \pm 0.009}$ | $0.41 \pm 0.01$ |
| WallRobot | $0.885 \pm 0.003$ | $\mathbf{0.640 \pm 0.003}$ | $0.907 \pm 0.005$ | $0.78 \pm 0.01$ | $0.776 \pm 0.007$ | $0.757 \pm 0.005$ |
| WhiteWine | $0.964 \pm 0.007$ | $0.878 \pm 0.009$ | $0.97 \pm 0.02$ | $\mathbf{0.850 \pm 0.005}$ | $\mathbf{0.848 \pm 0.007}$ | $0.85 \pm 0.01$ |
| Yeast | $0.98 \pm 0.02$ | $1.00 \pm 0.02$ | $0.99 \pm 0.03$ | $\mathbf{0.95 \pm 0.01}$ | $0.958 \pm 0.007$ | $0.97 \pm 0.03$ |
| Zoo | $\mathbf{0.19 \pm 0.03}$ | $\mathbf{0.16 \pm 0.02}$ | $0.20 \pm 0.02$ | $\mathbf{0.16 \pm 0.02}$ | $0.17 \pm 0.02$ | $\mathbf{0.16 \pm 0.01}$ |

Table 11: R2-score (for continuous targets) or accuracy (for categorical ones) of post-imputation regression or classification. Higher is better.

| Dataset | MICE | MissForest | GAIN | VAEAC | GSNN | NN |
|---------|------|------------|------|-------|------|-----|
| Boston | $\mathbf{0.57 \pm 0.08}$ | $\mathbf{0.6 \pm 0.1}$ | $\mathbf{0.50 \pm 0.10}$ | $\mathbf{0.5 \pm 0.1}$ | $\mathbf{0.5 \pm 0.1}$ | $\mathbf{0.50 \pm 0.09}$ |
| Breast | $\mathbf{0.96 \pm 0.02}$ | $0.95 \pm 0.02$ | $0.94 \pm 0.01$ | $0.95 \pm 0.02$ | $\mathbf{0.96 \pm 0.02}$ | $0.95 \pm 0.02$ |
| Concrete | $\mathbf{0.35 \pm 0.05}$ | $0.33 \pm 0.04$ | $0.28 \pm 0.06$ | $0.30 \pm 0.08$ | $0.32 \pm 0.05$ | $0 \pm 1$ |
| Diabetes | $\mathbf{0.37 \pm 0.06}$ | $0.34 \pm 0.06$ | $0.34 \pm 0.03$ | $0.34 \pm 0.04$ | $0.33 \pm 0.04$ | $0.27 \pm 0.06$ |
| Digits | $0.86 \pm 0.02$ | $0.887 \pm 0.008$ | $0.83 \pm 0.03$ | $0.892 \pm 0.010$ | $0.895 \pm 0.010$ | $\mathbf{0.912 \pm 0.010}$ |
| Glass | $0.44 \pm 0.08$ | $\mathbf{0.53 \pm 0.05}$ | $0.37 \pm 0.05$ | $0.49 \pm 0.09$ | $0.47 \pm 0.09$ | $0.48 \pm 0.09$ |
| Iris | $0.81 \pm 0.02$ | $\mathbf{0.84 \pm 0.02}$ | $0.66 \pm 0.06$ | $\mathbf{0.84 \pm 0.05}$ | $0.82 \pm 0.06$ | $0.73 \pm 0.09$ |
| Mushroom | $0.92 \pm 0.01$ | $0.972 \pm 0.003$ | $0.969 \pm 0.005$ | $\mathbf{0.987 \pm 0.001}$ | $0.986 \pm 0.002$ | $\mathbf{0.989 \pm 0.003}$ |
| Orthopedic | $\mathbf{0.71 \pm 0.02}$ | $\mathbf{0.72 \pm 0.03}$ | $0.60 \pm 0.03$ | $\mathbf{0.71 \pm 0.02}$ | $0.70 \pm 0.04$ | $0.61 \pm 0.04$ |
| Phishing | $\mathbf{0.75 \pm 0.02}$ | $0.73 \pm 0.03$ | $0.74 \pm 0.03$ | $\mathbf{0.75 \pm 0.01}$ | $0.74 \pm 0.04$ | $0.73 \pm 0.02$ |
| WallRobot | $0.55 \pm 0.01$ | $\mathbf{0.697 \pm 0.005}$ | $0.56 \pm 0.01$ | $0.62 \pm 0.02$ | $0.62 \pm 0.01$ | $0.64 \pm 0.02$ |
| WhiteWine | $0.13 \pm 0.02$ | $\mathbf{0.17 \pm 0.01}$ | $0.11 \pm 0.01$ | $\mathbf{0.18 \pm 0.02}$ | $0.17 \pm 0.01$ | $0.15 \pm 0.03$ |
| Yeast | $0.42 \pm 0.02$ | $0.41 \pm 0.02$ | $0.39 \pm 0.06$ | $0.42 \pm 0.01$ | $\mathbf{0.425 \pm 0.010}$ | $0.33 \pm 0.03$ |
| Zoo | $\mathbf{0.78 \pm 0.06}$ | $0.71 \pm 0.08$ | $0.67 \pm 0.06$ | $\mathbf{0.77 \pm 0.09}$ | $\mathbf{0.8 \pm 0.1}$ | $\mathbf{0.83 \pm 0.08}$ |

## D.2  MISSING FEATURES IMPUTATION

We evaluate the quality of imputations on different datasets (mostly from UCI (Lichman, 2013)). The evaluation is performed for VAEAC, GSNN (15) and NN (neural network; can be considered as a special case of GSNN where $p_\theta(z|x_{1-b}, b)$ is delta-function; produces single imputation). We compare these methods with MICE (Buuren & Groothuis-Oudshoorn, 2010), MissForest (Stekhoven & Bühlmann, 2011), and GAIN (Yoon et al., 2018).

We see that for some datasets MICE and MissForest outperform VAEAC, GSNN and NN. The reason is that for some datasets random forest is more natural structure than neural network.

The results also show that VAEAC, GSNN and NN show similar imputation performance in terms of NRMSE, PFC, post-imputation R2-score and accuracy. Given the result from appendix C we can take this as a weak evidence that the distribution of imputations has only one local maximum for datasets from (Lichman, 2013).

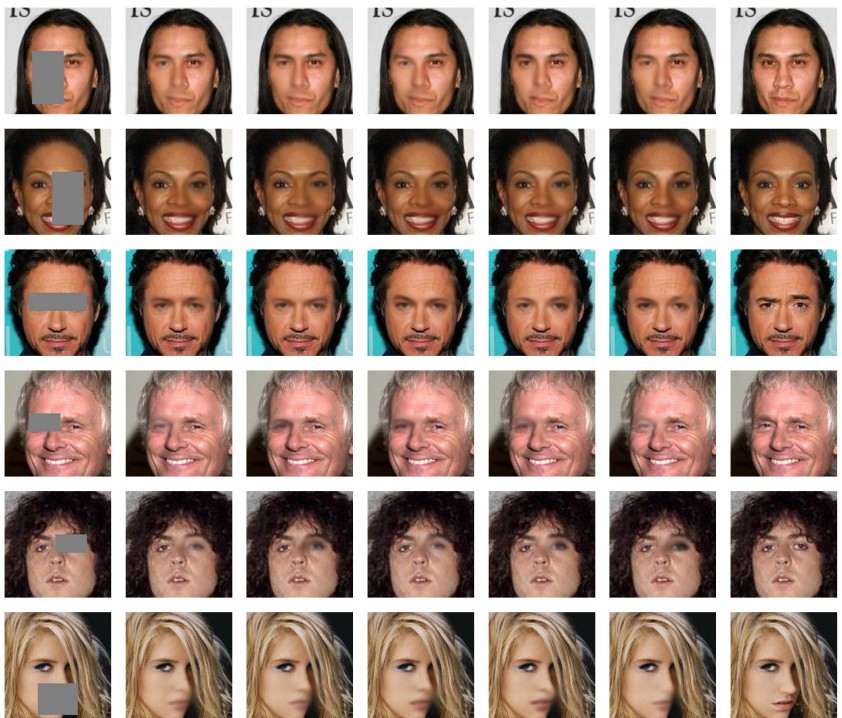

Figure 10: CelebA inpaintings with masks from (Li et al., 2017).
Left: input. The gray pixels are unobserved. Middle: samples from VAEAC. Right: ground truth.

### D.3 FACE INPAINTINGS

In figure 10 we provide samples of VAEAC on the CelebA dataset for the masks from (Li et al., 2017).

### D.4 GAIN FOR IMAGE INPAINTING

GAIN (Yoon et al., 2018) doesnt use unobserved data during training, which makes it easier to apply to the missing features imputation problem. Nevertheless, it turns into a disadvantage when the fully-observed training data is available but the missingness rate at the testing stage is high.

We consider the horizontal line mask for MNIST which is described in appendix A.3. We use the released GAIN code [5] with a different mask generator. The inpaintings from VAEAC which uses the unobserved pixels during training are available in figure 1. The inpaintings from GAIN which ignores unobserved pixels are provided in figure 11. As can be seen in figure 11, GAIN fails to learn conditional distribution for given mask distribution $p(b)$.

Nevertheless, we don't claim that GAIN is not suitable for image inpainting. As it was shown in the supplementary of (Yoon et al., 2018) and in the corresponding code, GAIN is able to learn conditional distributions when $p(b)$ is pixel-wise independent Bernoulli distribution with probability 0.5.

---

[5]https://github.com/jsyoon0823/GAIN

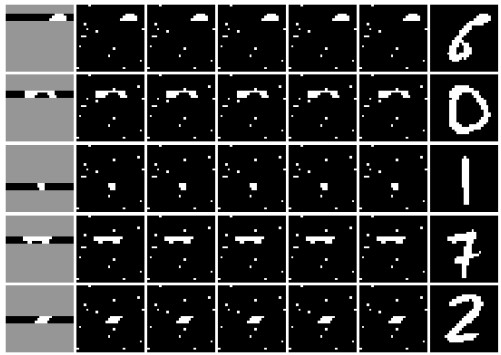

Figure 11: MNIST inpaintings from GAIN.
Left: input. The gray pixels are unobserved. Middle: samples from the model. Right: ground truth.

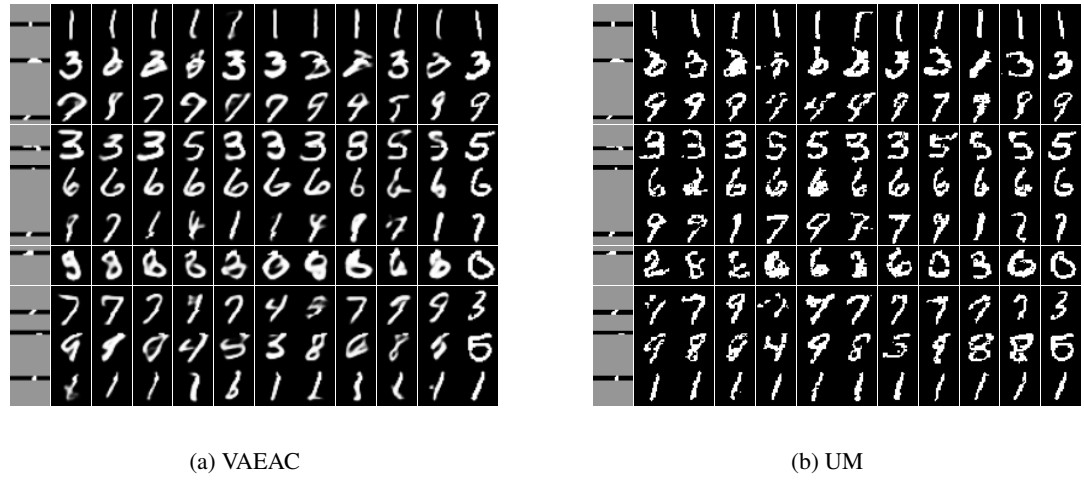

(a) VAEAC                                  (b) UM

Figure 12: MNIST inpaintings.
Left: input. The gray pixels are unobserved. Middle: samples from the model. Right: ground truth.

## D.5 UNIVERSAL MARGINALIZER: ILLUSTRATIONS

In figure 12 we provide samples of Universal Marginalizer (UM) and VAEAC for the same inputs.

Consider the case when UM marginal distributions are parametrized with Gaussians. The most simple example of a distribution, which UM cannot learn but VAEAC can, is given in figure 13.

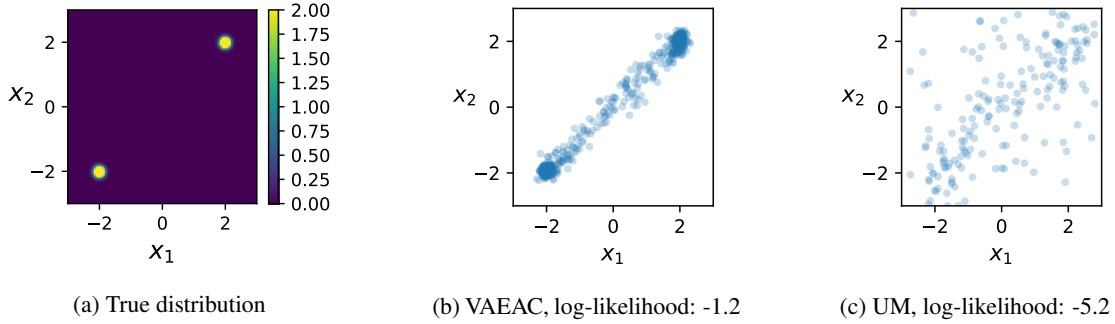

(a) True distribution        (b) VAEAC, log-likelihood: -1.2        (c) UM, log-likelihood: -5.2

Figure 13: Distribution learning: VAEAC vs UM.

