# OpenReview forum: "Variational Autoencoder with Arbitrary Conditioning"
_ICLR.cc/2019/Conference_

### Official Review · AnonReviewer1 · 2018-11-04
**Idea: Train a VAE to maximize the likelihood of a subset of the data while using the other subset for posterior inference. Poorly written paper with some interesting qualitative results.**

**Rating:** 6
**Confidence:** 3

**Review:**

The goal of this paper is to use deep generative models for missing data imputation. This paper proposes learning a latent variable deep generative model over every randomly sampled subset of observed features. First, a masking variable is sampled from a chosen prior distribution. The mask determines which features are observed. Then, the likelihood of the observed features is maximized via a lower bound. Inference in this latent variable model is achieved through the use of an inference network which conditions on the set of "missing" (to the generative model) features.

Novelty:
Generative models have a long history of being used to impute missing data. e.g. http://www.cs.toronto.edu/~fritz/absps/ranzato_cvpr2011.pdf, https://arxiv.org/pdf/1610.04167.pdf,
https://arxiv.org/pdf/1808.01684.pdf, https://arxiv.org/pdf/1401.4082.pdf [Appendix F]
It is a little difficult to guage what the novelty of this work is.

Clarity
This is a poorly written paper. Distilling the proposed methodology down to one paragraph was challenging since the text meanders through several concepts whose relevance to the overarching goal is questionable. For example, it is not clear what Section 3.2 adds to the discussion. The text describes a heuristic used in learning GSNNs only to say that the loss function used by GSNNs is not used in the experimental section for this paper -- this renders most of 4.3.2 redundant. There are issues like awkward grammar, sloppy notation, and spelling mistakes (please run spell check!) throughout the manuscript. Please use a different notation when referring to the variational distributions (do not re-use "p").

Experimental Results
The model is evaluated against MICE and MissForest on UCI datasets. RMSE and accuracy of classification (from imputed data is compared). The complexity of data considered is simplistic (and may not make use of the expressivity of the deep generative model). Why not run these experiments on datasets like MNIST and Omniglot?
Beyond that:
(a) was there any comparison to how classification performance behaves when using another neural network based imputation baseline (e.g. the method in Yoon et. al)?
(b) the *kind* of missingness considered here appears to be MCAR (the easiest kind to tackle) -- did you consider experiments with other kinds of missingess?

The qualitative results presented in this work are interesting. The method does appear to produce more diverse in-paintings than the method from Yeh et. al (though the examples considered are not aligned).

Table 5 claims negative log-likelihood numbers on MNIST as low as 61 and 41 (I assume nats...). These numbers do not make sense. How were they computed?


Priors on b:
What kind of priors on b did you experiment with?

---

> ### Author Response · Authors · 2018-11-26
> **Response to Reviewer 1 (Part 1/2)**
>
> We thank the reviewer for their feedback and comments. The reviewer raises several important concerns, which we address below.
>
> > It is a little difficult to guage what the novelty of this work is.
> The novelty of the paper is the model based on variational autoencoder which can be conditioned on the arbitrary subset of the features. The model generates unobserved features from joint conditional distribution over them using a single forward pass through neural network. To our best knowledge, such model has not been proposed and explored previously.
> In Susskind et al’11 (http://citeseerx.ist.psu.edu/viewdoc/download?doi=10.1.1.220.4604&rep=rep1&type=pdf ) authors train gated MRF at the lowest layer of DBN. Authors also propose an iterative process for filling-in missing pixels. We can claim that DBNs learning process is harder and less stable than the same for VAEs as we can see from experiments in the paper which use only 48x48 grayscale images. More importantly, the proposed method fills-in missing pixels in the single way while our paper propose a generative way to fill-in them in a number of significantly different ways.
> Rezende et al’14 (https://arxiv.org/pdf/1401.4082.pdf) require a VAE pretrained on the fully observed data and run MCMC process where each step is forward pass through VAE in order to get one conditional sample, which is computationally expensive.
> Sharir et al’16 (https://arxiv.org/pdf/1610.04167.pdf) also focuses mostly on the marginalization over the generated features for missing features imputation. The experiment in the paper consider only the classification quality under the missing data while in our work it is only a proxy metric which helps us to estimate the quality of imputations.
> Zhang et al’18 (https://arxiv.org/pdf/1808.01684.pdf) is only a preprint with no code released. From the practical point of view, the proposed method requires to solve a fixed-point problem which requires much more forward passes through neural networks than in our method. During training such problem has to be solved on every E-step of EM-algorithm. From the theoretical point of view the approximations like E_{p(y | x)} p(z | x, y) \approx p(z | x, E[y | x]) which are widely used in the paper do not sound reasonable, especially in the case when p(y | x) is multimodal which we address in our paper. We believe that in the case of multimodal dependencies such inaccurate approximations will prevent method from learning true joint conditional distribution.
>
> > For example, it is not clear what Section 3.2 adds to the discussion. The text describes a heuristic used in learning GSNNs only to say that the loss function used by GSNNs is not used in the experimental section for this paper -- this renders most of 4.3.2 redundant.
> Our paper is a follow-up of the Conditional Variational Autoencoder work, which claims that GSNN and hybrid model are important heuristics that increase model’s quality. Thus we think our theoretical and experimental evaluation of this technique is valuable for the community. Most of this evaluation is provided in the appendix, while the main paper just describes the summary of our study and contains a link to appendix for those who interested.
>
> > Please use a different notation when referring to the variational distributions (do not re-use "p").
> We would like to clarify that p(b | x) is not a variational distribution, but a user-defined distribution which describes the generative process of pairs (x, b) for the training stage: firstly x is sampled from the train set, then b is generated conditionally on x. Otherwise please describe the distribution you mean. We use only one variational distribution in the paper which we denote as q(z | x, b).
>
> > Why not run these experiments on datasets like MNIST and Omniglot?
> We find that MICE and MissForest are too computationally expensive to be used for MNIST and Omniglot. For instance, running MissForest on a dataset which contains less than 40000 objects and 60 features took about 10 days. MICE which produces 10 imputations for each object works even slower.

---

> > ### Author Response · Authors · 2018-11-26
> > **Response to Reviewer 1 (Part 2/2)**
> >
> > > Was there any comparison to how classification performance behaves when using another neural network based imputation baseline (e.g. the method in Yoon et. al)?
> > Thank you for this important question. We reimplemented their method based on the paper and the published code for MNIST inpainting. We ran the comparison on UCI datasets. We observe that GAIN works comparable with our method and for some datasets even worse than VAEAC. We can also notice that GAIN doesn’t use unobserved data during training, which makes it easier to apply to the missing features imputation problem. Nevertheless, when the fully-observed training data is available it turns into a disadvantage. For example, in inpainting setting GAIN cannot learn the conditional distribution over MNIST digits given one horizontal line of the image while VAEAC can. All these results are reported in the paper and in the supplementary.
> >
> > > The *kind* of missingness considered here appears to be MCAR (the easiest kind to tackle) -- did you consider experiments with other kinds of missingess?
> > For missing feature imputation we did not. Nevertheless, one can see in inpainting section, that model works well with highly structured kinds of missingness such as masks containing rectangles, horizontal lines and complex pattern structures (last two rows of figure 4).
> >
> > > Table 5 claims negative log-likelihood numbers on MNIST as low as 61 and 41 (I assume nats...). These numbers do not make sense. How were they computed?
> > It is the log-likelihood of the unobserved part of image given the observed part, estimated using importance sampling. Surely this log-likelihood is not comparable with log-likelihood of unconditioned generative models. The first reason is that the unobserved part contains 700 pixels instead of 784 in unconditional model. The second reason is that model uses the information from the observed part.
> >
> > > What kind of priors on b did you experiment with?
> > In missing features imputation we used component-wise Bernoulli distribution over b. In the inpainting section we also considered the equiprobable distribution over the masks which contains
> > + one missing rectangle
> > + one missing random half of the image
> > + missing center part of the image
> > + one observed horizontal line
> > + complex missing pattern proposed in the paper “Context Encoders”
> > In figure 4 we used the mixture of all those distributions except horizontal line at the training stage.
> > The samples from all used distributions are presented on pictures 1-4. The detailed description of those distributions over b are available in appendices A.2 and A.3.

---

> > > ### Comment · AnonReviewer1 · 2018-12-04
> > > **Clarified misunderstanding of the work, updated score, still borderline**
> > >
> > > My apologies for the delay. I'd like to acknowledge that I had a few misunderstandings that were clarified by the other reviews, the rebuttal and a few more passes through the latest manuscript -- thank you for that. I have updating my score to reflect this. I agree with R2 that the approach is novel but I am not yet convinced it is a good paper.
> > >
> > > There are a few axes of variation important to consider for proposed solutions to this problem:
> > > (a) Different kinds of missingness:
> > > MAR, MNAR, MCAR are three of the common kinds of missingness enountered in data. Currently, the manuscript does not experiment with the latter two kinds of missingness. The paper studies the MCAR case in Table 1. I'm left wondering what the table of results would look like when stratified by different kinds of missingness. Furthermore, for imputation with the goal of downstream classification, not all features of the data are created equal and only a small handful of them may be relevant for the classification task. How well does the proposed method reconstruct the most predictive subset of features (selected via some simple method such as L1 regularized regression)?
> > >
> > > (b) Multi-modality in the observed features:
> > > I think the paper does do a good job at tackling this axis of the problem by capturing multi-modality in the latent variable. This is where I see the utility of the qualitative experiments in image inpainting and the comparison to the Universal Marginalizer.
> > >
> > > (c) Computational complexity vs accuracy tradeoff:
> > > As pointed by external readers, there certainly are other deep generative model based methods for imputation. The authors point out the alternatives are slower *at test time*. This is true but it is also important to note that it is so because this approach amortizes that cost during *training* by attempting to condition on all possible missingness masks. It still remains to be seen whether this approach yields more accurate results than the other slower approaches.
> > >
> > > For reasons (a) and (c), I think this paper is borderline.

---

### Official Review · AnonReviewer3 · 2018-11-06
**interesting paper; missing/weak experiments**

**Rating:** 6
**Confidence:** 3

**Review:**

The paper presents a model for learning conditional distribution when arbitrary partitioning the input to observed and masked parts. The idea is to extend the conditional VAE framework such that the posterior is a function of an arbitrary subset of observed variables. Accordingly, reconstruction loss only penalizes the error in the reconstruction of masked (unobserved) variables. The method is compared against 1) classical approaches in missing data imputation on UCI benchmarks; 2) image inpainting against recently proposed GANS for the similar task, as well as; 3) against universal marginalizer, which learns conditional densities using a feedforward / autoregressive architecture.

My concern about the experimental results on missing data imputation is that strong competition such as Gondra et al’17 and Yoon et al’18 that report better results on UCI than classical approaches are not included. Could you please comment? See also [1,2] for other autoencoding architectures for this task.

While the derivation of the method is principled, it assumes that either the mask is known during the training OR one could efficiently sample a distribution of masks to learn arbitrary conditional densities. Given the exponential number of valid masks in a general setting, one only subsamples a small portion during the training. The question is whether the model can generalize well in this regime? The experimental results in this setting is not very encouraging, suggesting the proposed approach is effective only when the limitted mask patterns are known in advance.

[1] Gondara, Lovedeep, and Ke Wang. "Multiple imputation using deep denoising autoencoders." arXiv preprint arXiv:1705.02737 (2017).

[2] Zhang, Hongbao, Pengtao Xie, and Eric Xing. "Missing Value Imputation Based on Deep Generative Models." arXiv preprint arXiv:1808.01684 (2018).

---

> ### Author Response · Authors · 2018-11-26
> **Response to Reviewer 3**
>
> Thank you very much for taking the time to review our work.
>
> Thank you for pointing out the additional baselines for missing features imputation.
> Before commenting them we want to emphasize that our paper focuses on the learning joint conditional density of unobserved features. This problem has a lot in common with missing features imputation, but there are some differences. The majority of missing feature imputation methods do not consider the whole distribution over missing features. Usually, they restore the average or the most probable value of unobserved features, or the conditional marginal distribution for each unobserved feature. Such imputations can provide good RMSE metrics or post-imputation classification quality, but we consider them as not satisfactory for our problem setting.
> We comment the proposed baselines below one by one:
> 1. Gondara et al’17 focuses on missing features multiple imputation, but not on learning of the conditional distribution. They propose to train the same neural network starting from different initialization in order to obtain multiple imputations. Such approach is quite computationally expensive when multiple imputations are required. More importantly, there are no theoretical guarantees or experimental proof that such method will restore the true joint distribution of the unobserved features instead of the averages or marginals of the features.
> 2. Yoon et al’18 solves the same problem as we do. We reimplemented their method based on the paper and the code they published for MNIST inpainting. We ran the comparison on UCI datasets. We observe that GAIN works comparable with our method and for some datasets even worse than VAEAC. We can also notice that GAIN doesn’t use unobserved data during training, which makes it easier to apply to the missing features imputation problem. Nevertheless, when the fully-observed training data is available it turns into a disadvantage. For example, in inpainting setting GAIN cannot learn the conditional distribution over MNIST digits given one horizontal line of the image while VAEAC can. All these results are reported in the paper and in the supplementary.
> 3. Zhang et al’18 is hard to compare with, because the paper is only a preprint with no code released. From the practical point of view, the proposed method requires to solve a fixed-point problem which requires many more forward passes through neural networks than in our method. During training such problem has to be solved on every E-step of EM-algorithm. From the theoretical point of view the approximations like E_{p(y | x)} p(z | x, y) \approx p(z | x, E[y | x]) which are widely used in the paper do not sound reasonable, especially in the case when p(y | x) is multimodal which we address in our paper. We believe that in the case of multimodal dependencies such inaccurate approximations will prevent method from learning the true joint conditional distribution.
>
> > Given the exponential number of valid masks in a general setting, one only subsamples a small portion during the training. The question is whether the model can generalize well in this regime?
> In the general case when the distribution is arbitrary complex the answer is no. Nevertheless, if there are dependencies in the data which behaves similarly for all masks, including unseen ones, then the model would catch them. We can see exactly the described behavior on the 7th and 8th row on the figure 4. In this kind of masks each pixel has 80% probability to be missed, so there are at least 10^888 different most probable masks. Obviously, VAEAC never observes two equal masks in both training and testing stage. Nevertheless, as we see, it generalizes well for all masks generated from Bernoulli distribution.
> The problem with generalization appears when the dependencies in the data are completely different for various kinds of masks. For example, if we learn VAEAC using Bernoulli distributed masks, it learns only short-range dependencies in the data. If after that we close, for instance, right half of the image, then the model will produce unnatural inpaintings, because long-range dependencies are required for handling such kind of masks.
> Nevertheless, such cases of completely different dependencies induced by masks are rare. For instance, in our experiments the method shows to be tolerant to changing the scale of pattern mask from training to testing stage. The examples of pattern mask are presented on the last two rows of figure 4.

---

### Official Review · AnonReviewer2 · 2018-11-06
**Solid work on using VAEs for feature imputation**

**Rating:** 7
**Confidence:** 3

**Review:**

This paper introduces the VAEAC model, inspired by CVAEs, it allows conditioning on any subset
of the latent features. This provides a model able to achieve good results on image inpainting
and feature imputation tasks.

The paper appears to be technically sound, and the experiments are
thoughtfully designed. The writing is clear and the model is easy to
understand. The closest work to this of the Universal Marginalizer is
compared to well, with more compelling examples in the appendix. I
would have preferred if more of the experimental results were in the
main paper instead of in the appendix especially as the authors state
they chose to highlight their better results in the main paper.

While not the first model to try to handle modeling data with missing features, it is
still a fairly original and elegant formulation.

Minor details:

In equation (8) should x be x_b?

---

> ### Author Response · Authors · 2018-11-26
> **Response to Reviewer 2**
>
> Thank you for your kind review.
>
> In equation (8) the proposal network q is conditioned on the concatenation of observed and unobserved parts of the object x_{1 - b} and x_b which appears to be x.

---

### Public Comment · ~Anirudh_Goyal1 · 2018-11-26
**Missing Related work**

Hello,

There's a rich literature which allows to train generative models, as by training a transition operator of a markov chain, and hence thereby allowing missing features imputation.

Please find some of the references (work done by myself and few other colleagues).

1) Variational Walkback (https://arxiv.org/abs/1711.02282)
2) Infusion training, https://arxiv.org/abs/1703.06975
3) Non equilibrium Thermodynamics. https://arxiv.org/abs/1503.03585

Thanks! :)

---

> ### Author Response · Authors · 2018-11-26
> **RE: Missing Related work**
>
> Thank you for pointing out these references! We updated the related work section.
>
> Though these models solve unconditional data generation problem, they can be easily conditioned on the arbitrary subset of missing features.
> Nevertheless, these models use a framework of Markov chains which makes them more computationally expensive at the test time. They also require fully-observed training data to learn the transition operator.

---

> > ### Public Comment · ~Anirudh_Goyal1 · 2018-11-27
> > **Thanks!**
> >
> > Thanks for prompt reply. The intent was to just let the authors know, (and not get the citation) that one way we could get the missing features is by learning a transition operator, and they do solve conditional data generation problem too. But authors are right that these methods are computationally expensive.
> >
> > Thanks for citing these works, and being more thorough in related work section! :)

---

### Public Comment · ~Philip_Bachman1 · 2019-04-16
**Missing references to prior work**

I like this topic. I proposed solving imputation problems over arbitrary, random subsets of the input variables using a conditional VAE in my paper: "Data Generation as Sequential Decision Making" (Bachman et al., NIPS 2015) [1]. See Section 3 for the technical details matching those in Section 4 of your paper. Though, I did not work with inputs where some of the features were "absolutely missing", so the modifications you describe in Section 4.3.3 were not part of my paper.

In general, this class of problems can be thought of as a subset of orderless autoregression -- see the nice review paper "Neural Autoregressive Distribution Estimation" by Uria et al. (2016) [2] for more details. Section 4 is particularly pertinent. The concept of orderless autoregression has been around for a while, and it seems a lot of people are now "rediscovering" it, but without reference to prior work. See, e.g., the use of "masked language modelling" in BERT. This is precisely orderless autoregression, and it is not new. Of course, impressive new applications of the idea, e.g. BERT, are well worth publishing.

[1] -- https://arxiv.org/abs/1506.03504
[2] -- https://arxiv.org/abs/1605.02226

---

> ### Author Response · Authors · 2019-05-02
> **RE: Missing references to prior work**
>
> Thank you for pointing your work and NADE, which we will gladly cite in the next revision.
>
> Indeed, the objective in section 4.3.1 is a special case of your model. However, our model is simpler and likely faster since it does not require LSTM-based sequential decision making.
>
> We agree that the missing feature imputation problem can be solved as orderless autoregression. However, our approach shows that non-autoregressive imputation is also viable. We compare to an autoregressive Universal Marginalizer model in Section 5.3. We find that autoregressive model can perform better, but, unsurprisingly, is slow at test time.

---

### Meta-Review · Area_Chair1 · 2018-12-12
**Variational Autoencoder with Arbitrary Conditioning**

**Confidence:** 3
**Recommendation:** Accept (Poster)

**Metareview:**

This paper proposes a VAE model with arbitrary conditioning. It is a novel idea, and the model derivation and training approach are technically sound. Experiments are thoughtfully designed and include comparison with latest related works.

R1 and R3 suggested the original version of the paper was lack of comparison with relevant work and the authors provided new experiments in the revision. The rebuttal also addressed a few other concerns about the novelty and clarity raised by R3.

Based on the novel contribution in handling missing feature imputation with VAE, I would recommend to accept the paper. It is worth noticing that there is another submission to ICLR (https://openreview.net/forum?id=ByxLl309Ym) that shares a similar idea of constructing the inference network with binary masking, although it is designed for a pre-trained VAE model.

There are still two weaknesses pointed out by R3 that would help improve the paper by addressing them:
1. The paper does not handle different kinds of missingness beyond missing at random.
2. VAE model makes the trade-off between computational complexity and accuracy.
Point 1 would be a good direction for future research, and point 2 is a common problem for all VAE approaches. While the latter should not become a reason to reject the paper, I encourage the authors to take MCMC methods into account in the evaluation section.